# Cyanotoxins Increase Cytotoxicity and Promote Nonalcoholic Fatty Liver Disease Progression by Enhancing Cell Steatosis

**DOI:** 10.3390/toxins15070411

**Published:** 2023-06-25

**Authors:** Suryakant Niture, Sashi Gadi, Qi Qi, Leslimar Rios-Colon, Sabin Khatiwada, Reshan A. Fernando, Keith E. Levine, Deepak Kumar

**Affiliations:** 1Julius L. Chambers Biomedical Biotechnology Research Institute, North Carolina Central University, Durham, NC 27707, USA; 2NCCU-RTI Center for Applied Research in Environmental Sciences (CARES), RTI International, Durham, NC 27707, USA

**Keywords:** cyanotoxins, hepatotoxicity, unfolded protein response, steatosis, fibrosis, NAFLD

## Abstract

Freshwater prokaryotic cyanobacteria within harmful algal blooms produce cyanotoxins which are considered major pollutants in the aquatic system. Direct exposure to cyanotoxins through inhalation, skin contact, or ingestion of contaminated drinking water can target the liver and may cause hepatotoxicity. In the current study, we investigated the effect of low concentrations of cyanotoxins on cytotoxicity, inflammation, modulation of unfolded protein response (UPR), steatosis, and fibrosis signaling in human hepatocytes and liver cell models. Exposure to low concentrations of microcystin-LR (MC-LR), microcystin-RR (MC-RR), nodularin (NOD), and cylindrospermopsin (CYN) in human bipotent progenitor cell line HepaRG and hepatocellular carcinoma (HCC) cell lines HepG2 and SK-Hep1 resulted in increased cell toxicity. MC-LR, NOD, and CYN differentially regulated inflammatory signaling, activated UPR signaling and lipogenic gene expression, and induced cellular steatosis and fibrotic signaling in HCC cells. MC-LR, NOD, and CYN also regulated AKT/mTOR signaling and inhibited autophagy. Chronic exposure to MC-LR, NOD, and CYN upregulated the expression of lipogenic and fibrosis biomarkers. Moreover, RNA sequencing (RNA seq) data suggested that exposure of human hepatocytes, HepaRG, and HCC HepG2 cells to MC-LR and CYN modulated expression levels of several genes that regulate non-alcoholic fatty liver disease (NAFLD). Our data suggest that low concentrations of cyanotoxins can cause hepatotoxicity and cell steatosis and promote NAFLD progression.

## 1. Introduction

Climate change and global warming have led to a rise in the temperature of lakes and ponds conducive to the growth of algal blooms, particularly blue-green algae/cyanobacteria. Several factors like changes in salinity, higher carbon dioxide levels, changes in rainfall, increasing seawater levels, and coastal upwelling can promote further algal blooms growth (https://www.epa.gov/nutrientpollution/climate-change-and-harmful-algal-blooms; accessed on 13 March 2023) [1,2,3]. Some of the species of photosynthetic aquatic cyanobacteria produce a class of toxins called cyanotoxins in the aquatic system. These algal bloom/cyanobacteria produce cyanotoxins in large quantities which is a significant threat to the aquatic system’s irrigation and drinking water supplies, and are now considered a major class of environmental pollutants [1]. People may be exposed to cyanotoxins by drinking or bathing in contaminated water [1,4]. Human health data and water sampling results reported to the Center for Diseases Control (CDC’s-Waterborne Disease and Outbreak Surveillance System; WBDOSS) for the years 2009–2010 were associated with algal blooms. Cases were reported from New York, Ohio, and Washington, and all occurred in freshwater lakes and affected at least 61 persons. These 11 harmful algal-blooms-associated outbreaks represented 46% of the 24 outbreaks associated with untreated recreational water use reported from 2009 to 2010, and since 1978, 79% of the 14 freshwater harmful algal-blooms-associated outbreaks that have been reported to the CDC [5]. Recently, variable concentrations of cyanotoxins have been reported in water samples in different parts of the world which range from undetected levels to 7000 µg/L in 1 case and cylindrospermopsin (CYN) concentration in water of about 423.5 µg/g dry weight (DW) [6]. Other cyanotoxins/neurotoxins like BMAA (25.3 µg/L), DABA (21.1 µg/L), STX (24.2 µg/L), and ATX (35 µg/L) are found in water samples [6]. In fish, tissue accumulation of microcystins (MCs) varied as reported earlier [7]; for example, MC-LR levels ranged from 1.0 ± 1.4 µg/kg to 70 ± 5.0 µg/kg in *Pomoxis* freshwater fish [7]. It has been earlier reported that of fisherfolk who live near Meiliang Bay, Lake Taihu, 24/30 participants found MCs in their blood and another study indicated that chronic exposure to MCs could increase the risk of non-alcoholic fatty liver disease (NAFLD) [8,9]. The World Health Organization (WHO) suggested MC-LR drinking water standard value of 1 µg/L and a recreational water value of 10 µg/L (for a single variant) [10], and Health Canada suggested a standard value of 1.5 µg/L for drinking water for MC-LR [11].

Cyanotoxins have three routes of exposure to humans (1) contact, (2) ingestion, and (3) inhalation of aerosols. Importantly, the daily increase in algal blooms and the release of cyanotoxins into freshwater can threaten animals and humans through contaminated drinking water or the food chain [12,13]. Cyanotoxins cause several adverse health effects such as dermatologic, gastrointestinal, respiratory, and neurologic signs and symptoms. Cyanotoxins cause skin rash/irritation, eye irritation, gastrointestinal signs or symptoms, respiratory signs or symptoms, fever, headache, and neurological effects (https://www.cdc.gov/habs/materials/factsheet-cyanobacterial-habs.html; accessed on 16 March 2023). Cyanotoxins are divided into cyclic peptides, alkaloids, lipopeptides, non-protein amino acids, and lipoglycans. So far, 279 cyanotoxins have been identified [14]. Microcystin-LR (MC-LR) is the most studied and considered a significant algal bloom toxin globally [15]. MC-LR is one of the most potent toxins released from cyanobacteria and can cause different pathological conditions [16]. According to the International Agency for Research on Cancer (IARC 2010), MC-LR is classified as a Group 2B possible carcinogen, if humans are exposed to MC-LR chronically. MC-LR has a combination of leucine (L) and arginine (R) at positions 2 and 4 and, due to its unique cyclic structure, MC-LR has high stability and protection from heat and oxidation, and resistance to biodegradation (hydrolysis) [17]. Chronic low exposure to MC-LR can occur through drinking contaminated water, direct skin contact, or inhalation [18]. Because of its severe hepatotoxicity, MC-LR can cause liver diseases including liver cancer, and therefore MC-LR is the most studied microcystin [13]. The genus Microcystis contains several species including *M. aeruginosa* which are known to produce several microcystins (MCs) that enhanced toxicity in the kidney, liver, intestine, spleen, and other organs of several fish species [4], and recent reports suggest that cyanotoxins also affect several human organs such as the brain, stomach, small and large intestine, gut, lungs, kidneys, skin, and liver [18,19,20,21].

Importantly, numerous studies indicate that cyanotoxin exposure is not only linked with gastroenteritis but also with hepatic inflammation, liver diseases, and liver cancers [19,20,22,23]. A recent study demonstrated that MC-LR dysregulates lipid metabolism in mice fed with a high-fat diet (HFD). MC-LR induced liver damage by increasing AST and ALT levels, and lipid parameters in serum and hepatic steatosis by upregulation of PI3K/AKT/mTOR/SREBP1 signaling [24]. In zebrafish (Kras^V12^ transgenic a doxycycline-inducible HCC model) MC-LR (3 μg/L) increased HCC progression by downregulation of serine/threonine phosphatase 2A (PP2A) and by upregulation of β-Catenin in the Wnt signaling pathway [25]. Similarly, in the diethylnitrosamine (DEN)-induced HCC (two-stage carcinogenesis) model in rats, MC-LR acted as a new tumor promoter, and another cyanotoxin nodularin acted as a potent tumor promoter associated with a weak initiating activity. The study revealed that MC-LR is possibly carcinogenic to humans and nodularin is considered as not classifiable for HCC carcinogenicity [26]. Genotoxic activities of cyanobacterial toxin cylindrospermopsin (CYN) were studied in HCC HepG2 cells; exposure to CYN decreased cell viability, spheroid formation, deregulation of Phase I and II metabolic enzymes expression, and also affected the expression of genes associated with DNA-damage response [27]. Similarly, another study suggests that combined exposure to CYN/MC-LR induced DNA double-strand breaks, increased cell arrest in the G0/G1 phase, and increased the expression of *CYP1A1* and *CDKN1A*, *GADD45A* genes that are involved in DNA-damage response [28]. Acute intratracheal exposure to CYN in mice suggests that CYN accumulates in mice’s liver which leads to increased G and H, alveolar collapse, activated macrophages, increased levels of elastic/collagen fibers, polymorphonuclear (PMN) cells infiltration in the liver, and increased peroxidase activity in lung and hepatic tissues [29]. Cyanotoxins microcystin/nodularin (MC/NOD), cylindrospermopsin (CYN), and anabaenopeptin (AB) were detected in HCC patient serum, and the differential levels of MC/NOD/CYN were associated with HCC-related gene expression as well as in the regulation of PPAR signaling and lipid metabolism, suggesting that MC/NOD/CYN may modulate HCC pathogenesis by alteration of the lipid metabolism or by regulation of hepatic steatosis [30]. Importantly, a recent study established an association between average MC-LR water concentration and liver health status; the study indicated that MC-LR promoted HCC genesis in residents who drank reservoir water and were infected with HBV [31]. This evidence suggests that cyanotoxins enhance liver toxicity and promote liver diseases.

NAFLD is a global chronic liver disease considered a major public health burden [32]. NAFLD starts with hepatic steatosis and is characterized by triglyceride accumulation within hepatocytes [33,34], which subsequently progresses to hepatocellular carcinoma (HCC) via fibrosis and cirrhosis [33]. Studies indicated that cyanotoxins cause hepatotoxicity and exacerbate the risk of NAFLD [8,35,36,37]. Cyanotoxins such as microcystins, nodularin (NOD)*,* and cylindrospermopsin (CYN) can cause liver damage [38]. Cyanotoxin microcystin (MC) accumulates in hepatocytes by organic anion-transporting polypeptides (OATPS) [39]. Studies indicate that MC-LR inhibits Ser/Thr protein phosphatase 1 (PP1) and protein phosphatase 2A (PP2A) activity [40], which modulates signal transduction in cell-cycle regulation and tumor suppression [41]. Moreover, MC-LR-induced oxidative stress activates the NLRP3 (NOD-, LRR- and pyrin domain-containing protein 3) inflammasome leading to hepatic insulin resistance, metabolic abnormalities, and the risk of diabetes in NAFLD phenotypes [42]. Animal studies suggested that MC-LR induced cell steatosis in hepatocytes/liver tissues [43,44] by inhibiting fatty acid oxidation and by increasing hepatic inflammation [45]. Furthermore, MC-LR induced nonalcoholic steatohepatitis (NASH) in rats when rats were fed a high-fat or high-cholesterol diet [46]. Intraperitoneal (i.p.) injection of MC-LR and MC-RR in tilapia fish (*Oreochromis* sp.) affected the livers and kidneys. MC-RR increased ACP in the kidney and ALP in the liver. MC-LR causes microvesicular steatosis, megalocytosis and the necrotic process and MC-RR causes degenerative renal changes and glomerulopathy [47]. Exposure to low concentrations of nodularin (NOD) and MC-LR in rat hepatocytes increased 8-oxo-dG levels and induced oxidative DNA damage suggesting that longer exposure to NOD/MC-LR may be involved in hepatic tumor formation [48]. Combined oral exposure to cylindrospermopsin (CYN) and microcystins (MC) in rats showed necrotic hepatocytes in centrilobular areas and liver damage suggesting that MC/CYN induced genotoxic and liver damage in rats [49]. 

However, the molecular mechanism of how cyanotoxins increase hepatic signaling and the risk of NAFLD is poorly understood. In the current study, for the first time, we investigated the effects and impact of low concentrations of MC-LR, MC-RR, NOD, and CYN on hepatotoxicity, hepatic steatosis, and fibrosis in hepatocytes, HepaRG, and HCC cells. We also analyzed the impact of cyanotoxins (MC-LR and CYN) on global gene expression, including the gene that modulates the development of fatty liver disease.

## 2. Results

### 2.1. Cyanotoxins Decrease Cell Survival and Proliferation

In the current study, we used four cyanotoxins: MC-LR, MC-RR, NOD, and CYN. The structure of these cyanotoxins is presented in Appendix A. To understand the impact of low concentrations of cyanotoxins on liver cell metabolism and survival, we exposed terminally differentiated human bipotent progenitor liver cells, HepaRG cells (which maintain many characteristics of primary human hepatocytes [50]), and two HCC cell lines, HepG2 and SK-Hep1 cells, to increasing concentrations (1 to 500 nM) of MC-LR, MC-RR, NOD, and CYN for 72 h, and cell metabolic activities were analyzed by an MTT assay (Figure 1A). In HepaRG cells, treatment with MC-LR and MC-RR (1 to 250 nM) decreased cell survival significantly (~15 to 45%) and NOD and CYN also decreased cell survival at the 500 nM concentration (Figure 1A, left panels). However, increased metabolic activities were observed when HepaRG cells were exposed to 500 nM of MC-LR compared with a 250 nM concentration indicating that MC-LR modulates cytotoxicity differentially in HepaRG cells. Similarly, dose-dependent exposure to MC-LR, MC-RR, NOD, and CYN in HepG2 cells decreased cell survival (~20 to 50%) compared with vehicle-treated cells (Figure 1A, middle panels). Similar to HepaRG cells, exposure to MC-LR at a concentration of 500 nM increased the metabolic activities observed in HepG2 cells compared with a 250 nM concentration. However, no significant effect on cell survival was observed when SK-Hep1 cells were exposed to MC-LR, MC-RR, and NOD but reduced cell survival was observed when cells were exposed to CYN (Figure 1A, right panels) indicating that cyanotoxins increased cell cytotoxicity and reduced cell survival in HepaRG and HCC cells.

Further, we analyzed the impact of the real-time proliferation in HepaRG and HCC cells after exposure of the cells to cyanotoxins using an Incucyte (Sartorius) live cell imager. Exposure of HepaRG to MC-LR, MC-RR and NOD did not show any effect on cell proliferation; however, CYN exposure significantly reduced cell proliferation. Our data further suggest that exposure to MC-RR, NOD, and CYN in HepG2 cells and MC-RR in SK-Hep1 cells decreased cell proliferation significantly, indicating that cyanotoxins exert a cytotoxic effect in HCC cells and suppress cell proliferation (Figure 1B, upper and lower panels). Exposure to MC-LR at a concentration of 10 nM and 50 nM showed increased cell proliferation in HepG2 cells compared with vehicle-treated cells, and CYN at 10 nM showed increased cell proliferation in SK-Hep1 cells indicating that these cyanotoxins modulate proliferative activities differentially at low concentrations.

Since cyanotoxins induced cytotoxicity, we further examined the effects of cyanotoxins on the regulation of cell apoptosis markers by immunoblotting. Exposure to MC-LR and NOD (50 and 250 nM) in HepG2 cells showed a slight increase (not significant) in cleaved PARP expression and CYN did not show any effect on cleaved-PARP expression when cells were exposed for 72 h. (Figure 1C, left two panels). No significant decrease in pro-caspase 3 expressions was observed when the cell was exposed to MC-LR, NOD, or CYN (Figure 1C, left two panels). In SK-Hep1 cells, MC-LR, NOD or CYN at 10 to 250 nM concentrations did not show a significant effect on cleaved PARP or caspase 3 expression (Figure 1C, right two panels) suggesting that low concentrations of cyanotoxins do not induce cell apoptosis in HCC cells.

### 2.2. Cyanotoxins Regulate Inflammatory Signaling in HCC Cells

To address the cytotoxicity of cyanotoxins, we analyzed the expression of pro-inflammatory cytokines *IL-6* and *TNF-α* in HCC cells after treatment with 10–250 nM of cyanotoxins for 72 h. The RT/qPCR data suggest that exposure to MC-LR did not show any significant change in *IL-6* and *TNF-α* expression in HepG2 cells; however, the expression of *IL-6* and *TNF-α* decreased in SK-Hep1 cells compared with vehicle-treated cells. Exposure to NOD increased *IL-6* and *TNF-α* at 50 and 250 nM concentrations and *TNF-α* downstream oncogenic *TNFAIP8* transcript in HepG2 cells at 50 nM concentration (Figure 2A, right middle panel). However, a significant increase in *IL-6*, *TNF-α*, and *TNFAIP8* expression was observed in SK-Hep1 cells treated with CYN (Figure 2A, lower panel), suggesting that cyanotoxins regulate inflammatory signaling differentially in HCC cells.

We also analyzed inflammatory protein expression such as antioxidant-related superoxide dismutase1 (SOD1), catalase (CAT), IL-6, and TNFAIP8 after exposure to cyanotoxins for 72 h in HCC cells by immunoblotting. Our data suggested that low concentrations of MC-LR, NOD, and CYN (10–250 nM) increased slightly the expression of SOD1, CAT, and TNFAIP8 (not significant) in HepG2 and SK-Hep1-cells. No change in the expression of IL6 in HepG2 or SK-Hep1 cells was observed (Figure 2B, all panels). Further, we quantified endogenous reactive oxygen species (ROS) production using CellROX Green Reagent after exposing cells to cyanotoxins. Cell immunofluorescence intensity data suggest that cyanotoxins enhanced ROS production in HCC cells slightly but not significantly, at least with these concentrations of cyanotoxins (Figure 2C and Appendix A). Indeed, our data suggest that these cyanotoxins differentially regulate inflammatory signaling in HCC cells.

### 2.3. Cyanotoxins Activate UPR in HCC Cells

The unfolded protein response (UPR) signaling pathway is a protective cellular response activated by ER stress. Recent reports suggest that UPR activation can induce cell death upon persistent ER stress [51]. Since the liver is susceptible to ER stress and cyanotoxins modulate inflammatory signaling and cytotoxicity, we analyzed the impact of cyanotoxins in regulating UPR biomarkers. We exposed HCC cells to 10 to 250 nM of cyanotoxins, as indicated for 72 h, and the expression of UPR markers such as *IRE1a*, *eIF2a*, *ATF4*, *ATF6*, and *BIP* was analyzed by RT/qPCR (Figure 3A). Exposure to MC-LR in HCC cells did not show significant induction of UPR markers; however, NOD increased the expression of *IRE1a*, *ATF4*, and *ATF6* in HepG2 cells and *IRE1a* in SK-Hep1 cells. CYN increased the expression of *ATF4* in HepG2 cells and *ATF6* in SK-Hep1 cells (Figure 3A, all panels). In addition, the activation of UPR biomarkers was further analyzed by immunoblotting. As shown in Figure 3B, NOD increased the expression of BIP and ATF6 in HepG2 and SK-Hep1 cells and CYN increased pERK in SK-Hep1 cells (Figure 3B, lower panels). However, no statistically significant expression of UPR protein markers was observed when cells were exposed to cyanotoxins at these concentrations. Collectively the data indicate that NOD and CYN activate/regulate UPR signaling in HCC cells.

### 2.4. Cyanotoxins Induce Cell Steatosis in HCC Cells

Hepatic steatosis is an initial stage in the progression of nonalcoholic fatty liver disease (NAFLD) or alcoholic fatty liver disease (AFLD) and these stages can progress to HCC over time [52]. Several preclinical animal studies suggest that MC-LR alters hepatic inflammation and lipid content/steatosis in the liver [45,53,54]; however, the molecular mechanism is not well defined. Several lipogenic genes/proteins such as fatty acid synthase (*FASN*), acetyl-CoA carboxylase (*ACC*), stearoyl-CoA desaturase-1 (*SCD1*), fatty acid-binding protein-1 (*FABP1*), and the transcription factor sterol regulatory element-binding protein-1 (*SREBP1*) modulates lipogenic process/cell steatosis. Here we investigated the effect of MC-LR, NOD and CYN on lipogenic genes/proteins expression and cellular steatosis in the HCC cell models. Exposure to MC-LR (50 nM) significantly increased *FABP1* expression in HepG2 cells and *SREBP1* and *SCD1* in SK-Hep1 cells. Similarly, NOD induced *SREPB1*, *FABP1*, *SCD1*, *FASN*, and ACC in HepG2 cells and *SREPB1*, *SCD1*, *FASN*, and *ACC* in SK-Hep1 cells. On the contrary, CYN decreased lipogenic gene expression in both cell lines (Figure 4A, all panels). To analyze the role of cyanotoxins-mediated regulation of lipogenic protein expression in cell steatosis, we exposed HCC cells to MC-LR, NOD, and CYN (50 nM), and cells were further treated with oleic acid (OA) for 24 h. Oil Red O (ORO) staining demonstrated that MC-LR, NOD, and CYN significantly increased cell steatosis in HepG2 and SK-Hep1 cells compared with OA-treated cells (Figure 4B, upper and lower panels). Oil Red O (ORO) staining quantification demonstrated that cyanotoxins increased cell steatosis by 15–30% in both cell lines compared with OA treatments alone (Figure 4C, upper and lower panels). Collectively our data indicate that cyanotoxins induce cell steatosis in HCC cells.

To confirm the RT/qPCR results, we performed immunoblotting for SREPB1, FABP1, SCD1, FASN, ACC, and PPAR-α after exposure to cyanotoxins. Compared with vehicle treatment, CYN increased the expression of ACC in HepG2 significantly and ACC and SCD1 in SK-HEP cells, and NOD increased the expression of SCD1 in HepG2 and SK-HEP cells. No significant change in the expression of SREPB1, FABP1, FASN, and PPAR-α was observed when cells were exposed to MC-LR, NOD, or CYN (Figure 4D, left and right panels). To understand how cyanotoxins modulate cell steatosis in HCC cells we analyzed the effect of cyanotoxins on AKT/mTOR and cellular autophagy pathways since the activation of AKT/mTOR inhibits autophagy [55] and inhibition of autophagy increases cell steatosis in NAFLD [56]. Immunoblotting data demonstrated that CYN treatments increased only the phosphorylation of AKT (S473-AKT) and mTOR (S2448-mTOR) in HepG2 cells. MC-LR and NOD did not alter the phosphorylation status of AKT and mTOR. The expression of LC3B/II and other autophagic biomarkers such as Beclin1 and 4EBP1 did not change after exposure to cyanotoxins. Interestingly, p62 expression was increased slightly in SK-Hep1 cells after NOD and CYN exposure but not significantly (Figure 4E, upper and lower panels), suggesting that cyanotoxins perhaps enhanced steatosis by impairing the autophagic function in HCC cells.

### 2.5. Cyanotoxins Regulate Fibrosis in HCC Cells

In NAFLD, non-alcoholic steatohepatitis (NASH) is a severe form of fatty liver disease that can modulate liver fibrosis, cirrhosis, and HCC [57]. Our data indicated that cyanotoxins treatments enhance cell steatosis, an initial stage of NAFLD progression. Here we further analyzed the impact of cyanotoxins on cell fibrogenic signaling in HCC cells. HepG2 and SK- Hep1 cells were exposed to 10–250 nM of MC-LR, NOD, and CYN for 72 h, and the expression of *p21*, *MMP2*, *TIMP2*, *TGFB1*, *FGF-23*, and *Cytokeratin7* (a fibrogenic gene biomarker) was analyzed by RT/qPCR (Figure 5A). Exposure to MC-LR significantly increased mRNA expression of *TGFB1*(~3.1-fold), *p21* (~1.3-fold), *CX3CR1* (~4.7-fold), and *FGF-23* (~2.45-fold) in HepG2 cells compared with vehicle-treated cells. *TIMP2*, *AST*, and *Cytokeratin 7* expressions also increased in SK-Hep1 cells after MC-LR treatments (Figure 5A, upper panels). However, different concentrations of NOD treatments induced *FGF23* and *CX3CR1* in HepG2 cells and *FGF23* and *CX3CR1* in SK-Hep1 cells (Figure 5A, lower panels). In addition, we analyzed the expression of the four fibrosis-signaling protein markers (TGF-β, MMP2, TIMP2, and p21) by Western blotting after exposing the HCC cells to MC-LR, NOD, and CYN as indicated (Figure 5B). Western blotting data indicated that compared with vehicle treatment, MC-LR increased the expression of TIMP2 and p21 HepG2 cells, and NOD slightly increased (not significantly) the expression of MMP2 and TIMP2 in SK-Hep1 cells (Figure 5B, right and left panels). Additionally, in another experiment, HCC cells were exposed to CCl_4_ (fibrosis inducer agents) alone or in combination with cyanotoxins as indicated (Figure 5B). Interestingly, compared with CCl_4_-treated cells, the expression of TGFβ, TIMP2, MMP2, and p21 was increased slightly in HepG2 cells when pre-treated with MC-LR and NOD (Figure 5B, left two panels) and MMP2 in SK-Hep1 cells when pre-treated with MC-LR and NOD (Figure 5B, right two panels). Although the expression of fibrosis markers is not significantly modulated by cyanotoxin expression, our data suggest that cyanotoxins may activate fibrogenic signaling in HepG2 and SK-Hep1 cells.

### 2.6. Chronic Exposure of MC-LR to HCC HepG2 and SK-Hep1 Cells Induces Lipogenic and Fibrosis Biomarker Expression

To investigate the significance of MC-LR in NAFLD, we exposed HCC HepG2 and SK-Hep1 cells to MC-LR at 10 and 50 nM concentrations for 30 days, and the expression of lipogenic biomarkers was analyzed by RT/qPCR and immunoblotting (Figure 6A,B). Chronic exposure to MC-LR did not show a significant effect on lipogenic *SREPB1*, *SCD1*, *FASN*, and ACC gene expression in SK-Hep1 cells; however, upregulation of *SREPB1*, *SCD1*, and *FABP1* was observed in HepG2 cells (Figure 6A, lower panel). Interestingly, exposure to MC-LR increased ACC, FASN, and SCD1 proteins in SK-Hep1 cells (although not significantly); however, compared with vehicle-treated HepG2 cells, ACC protein expression was increased in HepG2 cells and increased expression of SCD1 and SREBP1 was also observed slightly when cells were exposed to MC-LR for 30 days (Figure 6B, left and right panels) suggesting that chronic exposure to MC-LR regulates lipogenesis in HCC cells.

We further analyzed the effect of chronic MC-LR (30 days) exposure on fibrosis biomarkers. RT/qPCR data suggest that exposure to MC-LR at 10 and 50 nM concentrations induced the expression of *TIMP2* significantly and upregulation of *TGFB1*, *FGF-23,* and *CX3CR1* was observed in SK-Hep1 cells. In addition, the expression of *FGF-23, CX3CR1,* and *AST* also increased significantly after MC-LR exposure to HepG2 cells for 30 days (Figure 6C, upper and lower panels). Western blotting data further indicate that exposure to MC-LR for 30 days slightly increased the expression of TGF-β1, MMP2, and p21 in SK-Hep1 cells and TGF-β and MMP2 in HepG2 cells compared with vehicle-treated cells (Figure 6D, left and right panels). Indeed, the data suggest that chronic exposure to MC-LR not only increased lipogenic signaling but also upregulated fibrosis signaling in HCC cells.

### 2.7. Cyanotoxin Exposure Modulates NAFLD-Related Gene Expression

To understand the biological significance of how physiologically relevant low concentrations of MC-LR and CYN modulate NAFLD and other pathways, we used three liver cell models: (1) human hepatocytes, (2) hepatic cell line HepaRG, and (3) HCC HepG2 cells. These cells were exposed to MC-LR and CYN in triplicates for 72 h and the impact of MC-LR and CYN on global gene expression was analyzed by RNA-Seq. Heatmaps and volcano plotting revealed that with exposure to MC-LR in human hepatocytes, 3803 genes were upregulated and 4092 genes were downregulated (*p* < 0.05), whereas with exposure to CYN, 4760 genes were upregulated and 4915 genes were downregulated (*p* < 0.05). In HepaRG cells exposed to MC-LR, 170 genes were upregulated and 403 genes were downregulated (*p* < 0.05); exposure to CYN resulted in the upregulation of 1999 genes and the downregulation of 1,829 genes (*p* < 0.05). Similarly, 306 genes were upregulated and, 418 genes were downregulated when HepG2 cells were exposed to MC-LR, and 185 genes were upregulated and 273 genes were downregulated when cells were exposed to CYN (Figure 7A,B, left panels). These data suggest that cyanotoxins affect or alter global gene expression in human primary hepatocytes rather than immortalized human hepatic cell line HepaRG or HCC HepG2 cells.

Interestingly, GeneRatio/gene ontology analysis revealed that cyanotoxins upregulate several common pathways irrespective of cell lines used in this study. For example, MC-LR or CYN exposure upregulates cellular pathways/processes such as oxidative phosphorylation, carbon metabolism, ribosomal, Parkinson’s disease, and Huntington’s disease in hepatocytes. Interestingly, MC-LR or CYN upregulates the NAFLD pathway in hepatocytes (Figure 7A,B upper right panels). In HepaRG cells, MC-LR or CYN upregulates oxidative phosphorylation, ribosomal, Parkinson’s disease, Alzheimer’s disease, and mTOR as well as the NAFLD pathway (Figure 7A,B middle right panels). In HCC HepG2 cells, MC-LR upregulates oxidative phosphorylation, ribosomal, Parkinson’s disease, Huntington’s disease and Alzheimer’s disease as well as the NAFLD pathway (Figure 7A,B lower right panels). The top-upregulated and down-regulated 10 genes in hepatocytes, HepaRG cells, and HepG2 cells after MC-LR and CYN exposure are presented in Appendix A.

Importantly, RNA-Seq data suggest that MC-LR and CYN exposure shows upregulation of the NAFLD pathway in hepatocytes, HepaRG, and HepG2 cells, suggesting that cyanotoxins modulate NAFLD progression (Figure 7A,B). Further, we analyzed the RNA-seq data and identified key genes that contribute to NAFLD progression directly or indirectly. RNA-seq data revealed that MC-LR exposure upregulates *AVP*, *TEC*, *PRAMEF14*, *SLAMF9, TTC-36*, *GPR52*, *CAPZA3*, *LECT2*, and *GSC* genes in hepatocytes (Figure 8A), whereas MC-LR exposure downregulates *BMP6*, *PTGS2*, *TEK*, *FOXL1*, *PDLIM4*, *SIK1*, *DLAGP5*, *LHX2*, *TFF2*, and *TFF1* genes. (Figure 8A). CYN exposure to hepatocytes shows only *LECT2* and *GCS* gene upregulation and *PTGS2*, *FOXL1*, *TFF2*, and *TFF1* downregulation (Figure 8A). We performed a literature search, and the roles of these genes in liver diseases is presented (Figure 8A). We validated the RNA-seq data by RT/qPCR, and the expression of top-upregulated and downregulated genes in hepatocytes after exposure to MC-LR and CYN was analyzed (Figure 8A, lower panels). MC-LR exposure upregulated *TEC*, *GPR52*, *GSC*, *BMP6*, *PTGS2*, *TEK* and CYN exposure upregulated *PRAMEF14*, *SLAMF9*, *GPR52*, *CAPZA3*, *GSC*, *BMP6*, *PTGS2*, *SIK1*, *TFF1* genes in hepatocytes (Figure 8A, lower panels). However, in HepaRG cells MC-LR downregulated *PRAMEF14* and *GPR52* genes. CYN exposure to HepaRG downregulated *CAPZA3*, *LECT2*, *TEK*, and *PDLIM4* genes and upregulated the *SIK1* gene (Figure 8B, left and right panels). In HCC HepG2 cells *PRAMEF 14*, *SLAMF 9*, *TTC-36*, *LECT2,* and *PDLIM4* genes were upregulated after MC-LR treatment, whereas CYN exposure downregulated *PRAMEF14*, *GPR52*, *CAPZA3*, *LECT2*, *CGS*, *TPRM1*, *TEK*, *PDLIM2*, and *TFF1* in HepG2 cells (Figure 8C, left and right panels). Indeed, our data suggest that exposure to cyanotoxins modulated the expression of a set of genes that participate in fatty liver diseases or NAFLD development.

## 3. Discussion

In the current study, we used liver cell models to investigate the biological effects of MC-LR, MC-RR, CYN, and NOD, using physiologically relevant concentrations on liver cell toxicity and NAFLD modulation. Our data suggest that exposure to MC-LR, MC-RR, CYN, and NOD in HepaRG and HepG2 cells decreased cell metabolic activities. MC-RR, CYN, and NOD inhibited real-time cell proliferation in HCC cells. NOD and CYN increased pro-inflammatory IL-6 and TNFa expression and modulated SOD1, CAT, and TNFAIP8 protein expression. However, no significant ROS production was observed when HCC cells were exposed to cyanotoxins in vitro, at least at low concentrations of cyanotoxins. An earlier study indicated that even at 2 and 10 ng/mL concentrations, MC-LR and NOD-induced ROS production in hepatocytes can contribute to HCC progression [48]. Chronic exposure to MC-LR (83 days) induced ROS production in HCC HepG2 cells as reported earlier [58]. Since cyanotoxins modulate ROS production and ROS accumulation can activate UPR signaling, we analyzed the impact of MC-LR, NOD, and CYN on the expression of UPR biomarkers. Our data suggest that exposure to NOD and CYN activates UPR biomarkers such as IREa ATF6, ATF4, pERK, and BIP in HCC cells. Interestingly, the earlier study suggests that, in the zebrafish model, MC-LR-mediated ER stress could be protected when male zebrafish were intraperitoneally injected with N-acetylcysteine (NAC), suggesting that cyanotoxins activate UPR signaling [59].

Since cyanotoxins dysregulate the metabolic and UPR signaling, in the current study we further investigated the impact of cyanotoxins on cell steatosis and an early event of NAFLD and alcoholic fatty liver disease (AFLD). NAFLD is a common chronic liver disease and it is an early-stage NAFLD where lipid accumulation in the hepatocytes can occur that triggers subsequent pathologies of the liver such as liver fibrosis, cirrhosis, and HCC [60,61,62]. Our data suggest that MC-LR, NOD, and CYN exposure increased oleic acid (OA)-mediated induction of cell steatosis in HCC cells. This accumulation of lipid droplets in liver cells is due to the upregulation of the lipogenic genes and protein expression when cells are exposed to cyanotoxins. Earlier studies also indicated that MC-LR increased cell steatosis in hepatocytes and liver tissues [43,44]. Mechanistically, MC-LR inhibited fatty acid oxidation and increased hepatic inflammation [45], and induced nonalcoholic steatohepatitis (NASH), when rats were fed with a high-fat or high-cholesterol diet [46]. Moreover, functional autophagy (lipophagy) clears the accumulation of lipid droplets and reduces lipotoxicity in the liver by enhancing lipid metabolism [56,63,64], and our data suggest that cyanotoxins activate AKT/mTOR signaling partially, suppress autophagy by increasing the expression of p62, and increase cell steatosis in HCC cells. In addition, our RT/qPCR and immunoblotting data suggest that cyanotoxins also modulate fibrosis signaling in HCC cells. MC-LR significantly increased *TGF-β1*, *p21*, *cytokeratin 7*, and *FGF-23* in HepG2 cells and *TIMP2*, *AST*, and *cytokeratin 7* expressions in SK-Hep1 cells. In addition, chronic exposure to MC-LR in HCC cells upregulated the expression of TGF-β1, MMP2, and TIMP2. Upregulation of fibrosis markers was observed when cells were exposed to cyanotoxins and cyanotoxins plus CCl_4_. Similarly, Gu et al. recently showed that exposure to MC-LR (15 or 30 μg/kg) significantly induced liver fibrosis in mice liver [65]. Collectively our data and recent studies confirm that cyanotoxins enhance cell steatosis and fibrosis signaling in liver cell models.

We performed RNA-seq and analyzed the global gene expression to understand the biological significance of cyanotoxins. Gene ontology analysis suggests that MC-LR or CYN exposure upregulates several pathways related to Parkinson’s disease, Huntington’s disease, Alzheimer’s disease, and the NAFLD pathway. We further analyzed the key genes modulated by NAFLD in human hepatocytes and liver cell lines. RNA-seq data suggest that MC-LR upregulates *PRAMEF14*, *AVP*, *TEC*, *SLAMF9*, *TTC-36*, *GPR52*, *CAPZA3*, *LECT2*, and *GSC* genes whereas MC-LR downregulates *BMP6*, *PTGS2*, *TEK*, *FOXL1*, *PDLIM4*, *SIK1*, *DLAGP5*, *LHX2*, *TFF2*, and *TFF1* genes in hepatocytes. CYN exposure to hepatocytes shows only *LECT2* and *GSC* gene upregulation and *PTGS2*, *FOXL1*, *TFF2*, and *TFF1* downregulation. The literature search revealed that these genes are involved in liver diseases and other several diseases. For example, *AVP* is involved in the differentiation of cardiac fibroblasts into myofibroblasts by modulation of the pro-fibrosis effect and by modulation of the expression of transforming growth factor-β (*TGF-β*) and collagen [66]. The *AVP* deficiency in CCl_4_-induced cirrhotic hamster livers showed a significant decrease in alkaline phosphatase (ALP) levels in serum, increased matrix metalloproteinase-13 (MMP-13), and decreased the expression of tissue inhibitor of metalloproteinase-2 (TIMP-2). The study revealed that *AVP* deficiency cleared the signs of liver histopathological reversion and diminution in type I collagen deposits in the liver [67]. The loss of non-classical *SLAM* family receptors *SLAMF8* and *SLAMF9* significantly protected against lipopolysaccharide (LPS)-induced liver injury in mice [68]. Higher expression of the gamma-glutamyl transpeptidase-to-platelet ratio (*GPR*) was observed in chronic hepatitis B (CHB) and non-alcoholic fatty liver disease (NAFLD) patients and GPR could be used as a non-invasive marker to predict liver fibrosis and cirrhosis in CHB-NAFLD individuals [69]. On the other hand, MC-LR exposure in hepatocytes downregulated several gene expressions, for example, in NAFLD bone morphogenetic protein 6 (BMP6) expression was up-regulated and associated with hepatic steatosis [70]. The study further pointed out that BMP6-induced cell steatosis in hepatocytes is not associated with hepatic inflammation [70]. The deficiency LIM homeobox gene *LHX2* developed liver fibrosis in mice and LHX2 negatively regulated hepatic stellate cell (HSC) activation [71]. Interestingly, the winged helix transcription factor *FOXL1* is expressed in rare cells in the normal liver but was dramatically induced in mice livers that had undergone bile duct ligation or a choline-deficient, ethionine-supplemented diet, suggesting that Foxl1 is a marker of bipotential hepatic progenitor cells [72]. These studies indicate that MC-LR modulates several key genes that participate in the development of fatty liver disease. The impact of exposure to these cyanotoxins at physiologically relevant concentrations in the development of liver disease in vivo is currently being studied in our laboratory.

## 4. Conclusions

Our data suggest cyanotoxins even at low and physiologically relevant concentrations could induce hepatotoxicity, and decrease cell survival and proliferation in liver cell models. Although these cyanotoxins did not significantly increase reactive oxygen species intracellularly, NOD and CYN modulated pro-inflammatory gene expression and UPR protein marker expression. Our data suggest that MC-LR, CYN and NOD acute/chronic exposure increased cell steatosis and fibrosis in HCC cells by modulation of lipogenic gene expression, and fibrosis biomarker expression. RNA seq data suggested that MC-LR and CYN induced NAFLD-related gene expression that could promote non-alcoholic fatty liver disease development and progression in human hepatocytes and HCC.

## 5. Materials and Methods

### 5.1. Cell Culture and Cyanotoxins Preparation

Cryopreserved human hepatocytes (HPCH05+), and a hepatocyte thawing and plating medium were obtained from Xenotech (Kansas City, KS, USA). Human bipotent progenitor cell line HepaRG (Cat # HPRGC10) was purchased from ThermoFisher Scientific (Waltham, MA, USA). HepaRG cells are capable of differentiating into two different cell phenotypes (i.e., biliary-like and hepatocyte-like cells) as described earlier [50]. HepaRG cells were cultured in William’s E Medium (Cat # 12551032) containing 1% GlutaMax (Cat # 35050061). Hepatocellular carcinoma (HCC) HepG2 (Cat # HB-8065) and SK-Hep1(Cat # HTB-52) cells were purchased from *American Type Culture Collection (ATCC;* Manassas, VA, USA). Cells were grown in Dulbecco’s Modified Eagle *Medium* (DMEM; Invitrogen, Carlsbad, CA, USA) supplemented with 5% fetal bovine serum (FBS, Access Biologicals, Vista, CA, USA) and penicillin/streptomycin (50 U/mL). Cells were incubated in a cell culture incubator at 37 °C supplied with 5% CO_2_. When cells reached 70–80% of the confluence they were used in experiments. We also exposed HepG2 and SK-Hep1cells to MC-LR (10 and 50 nM) for 30 days and the expression of lipogenic and fibrosis biomarkers was analyzed. 

We purchased microcystins from Enzo Life Science (Farmingdale, NY, USA). Microcystin-LR (Cat # ALX-350-012-C100) was dissolved in DMSO, Microcystin-RR (Cat # ALX-350-043-C100) was dissolved in 80% methanol, nodularin (NOD) (Cat # ALX-350-061-C100) dissolved in methanol: water (1:1) and cylindrospermopsin (CYN) (Cat # ALX-350-149-C100) in methanol (100%). Carbon tetrachloride (CCl_4_, ≥ 99.5%, Cat # 289116) was obtained from Sigma-Aldrich (St. Louis, MO, USA) and dissolved in EtOH (99.5%).

### 5.2. MTT Assay

HepaRG and HCC HepG2 and SK-Hep1Cells (1 × 10^4^ cells/well) were grown in 96 well plates and treated with vehicle and/or increasing concentrations (1 to 500 nM) of MC-LR, MC-RR, NOD, and CYN for 72 h. Cells were then incubated with MTT (3-(4,5-dimethylthiazol-2-yl)-2,5-diphenyltetrazolium bromide) reagent (5 µL/well; Stock 5 mg/mL in PBS) for 1 h at 37 °C. Cells were carefully washed with PBS, and formazan crystals were dissolved by adding 100 µL DMSO in each well. Cell survival was determined by reading the plates at 570 nm using a Fluostar Omega plate reader (BMG Lab tech, Cary, NC, USA). The experiments were repeated three times.

### 5.3. ROS Quantification

HepG2 and SK-Hep-1 cells were cultured in 96 well plates in triplicates (5000 cells/well) for 12h and cells were then treated with vehicle or 10 and 50 nM of MC-LR, NOD, and CYN for 72 h. After 72 h, cells were treated with CellRox green reagent (ThermoFisher Cat # C10444) and incubated at 37 °C for 30 min. After washing with PBS, cells were observed under a Keyence BZX-810 fluorescence microscope 10× objective (Keyence Itasca, IL, USA) and photographed. In other experiments, the effect of cyanotoxins on endogenous ROS production was quantified using FLUOstar^®^ Omega plate reader (BMG Lab tech, Cary, NC, USA) with excitation/emission at 485/520 nm, and plotted.

### 5.4. RT/qPCR

HepG2 and SK-Hep1 cells were treated with vehicle or different cyanotoxins as indicated for 72 h, and total RNAs from HepG2 and SK-Hep1 cells were isolated using TRIZOL reagent (Life Technologies, Carlsbad, CA, USA). One microgram (equal amounts) of RNAs was reverse transcribed using the High-Capacity cDNA Reverse Transcription kit. Then, cDNA was incubated with a Power SYBR Green PCR master mix (Applied Biosystems; Foster City, CA, USA) with indicated gene-specific forward and reverse primers (Appendix A). *GAPDH* forward and reverse primers were used to assess *GAPDH* gene expression as an internal control in the PCR reaction. A QuantStudio-3 PCR system (Applied Biosystems; Foster City, CA, USA), was used to run all PCR reactions and analysis of gene expression as per the manufacturer’s protocols.

### 5.5. Western Blotting

HepG2 and SK-Hep1 cells were treated with vehicle or different cyanotoxins as indicated for 72 h. Cells were washed with cold PBS and lysed in cell lysis buffer (Cell Signaling Technology, Danvers, MA, USA) containing a protease inhibitor cocktail (Roche, Indianapolis, IN, USA). Cell lysates were prepared by centrifugation at 10,000 RPM for 15 min, and the supernatants were used for protein qualification. A Bio-Rad protein assay reagent (Bio-Rad, Hercules, CA, USA) was used for the determination of protein concentrations. Immunoblotting was performed as described previously [73]. Briefly, sixty micrograms of protein lysates were electrophoresed by using NuPAGE 4-12% Bis-Tris-SDS gels (Invitrogen). Proteins were transferred to polyvinylidene difluoride (PVDF) membranes (Millipore, Billerica, MA, USA). The membranes were washed with 1X Tris-buffered saline with 0.1% Tween 20 (TBS-T) and blocked in 1X blocking buffer (Sigma-Aldrich, St. Louis, MO, USA) for 1 h. The membranes were incubated with primary antibodies (1:1000 dilution) overnight at 4 °C as per the manufacturer’s protocols. The following antibodies were obtained from Cell Signaling Technology (Danvers, MA, USA): anti-PARP (Cat # 9542S), anti-caspase 3 (Cat # 9662S), anti-cleaved caspase 3 (Cat # 9664S), anti-β-actin (Cat # 4970S), anti-SOD1 (Cat # 2770S), anti-CAT (Cat # 12980T), anti-IL-6 (Cat # 12153), anti-β-tubulin (Cat # 2128S), anti-pERK (Cat # 5683P), anti-BIP (Cat # 3177P), anti-p-eIF2α (Cat # 3398P), anti-pS2448-mTOR (Cat # 2971S), anti-mTOR (Cat # 2983S), anti-pS473-AKT (Cat # 3787S), anti-AKT (Cat # 9272S), anti- LC3B (Cat # 4108S), anti-p62 (Cat # 5114S), anti-Beclin-1 (Cat # 3495S), anti-4EBP1 (Cat # 9452S) anti-p21(Cat # 2947S), anti-TIMP2 (Cat # 5738S), anti-MMP2 (Cat # 13132S), anti-TGFβ (Cat # 3711S), anti-fatty-acid synthase (FASN) (Cat # 3180S), anti-SCD1 (Cat # 2794S), anti-ACC (Cat # 3662S). We also obtained: anti-L-FABP (Cat # ab7366) from Abcam; anti-PPARa (Cat # 15540-1-AP) and anti-TNFAIP8 (Cat # 15790-1-AP) antibodies from Proteintech (Rosemont, IL, USA); anti-SREBP1 (Cat # sc-13551) anti-ATF6 (SC-22799) from Santa Cruz Biotechnology (Dallas, TX, USA). After overnight incubation, the Western blots were washed three times with TBST (5 min each) and then incubated in the appropriate (anti-rabbit or anti-mouse) secondary antibody (1:10,000 dilution) (Jackson ImmunoResearch, PA, USA) for 1h at room temperature. The immunoblots were developed using ECL solution (ThermoScientific, Cat # 34580) and blots were visualized using the Azure instrument (C-500 Bio-system). Immunoblots were repeated three times and one set of data was presented. Immunoblot band intensities were quantified by Image J (https://imagej.nih.gov/ij/; Version 1.53t; accessed on 25 May 2023).

### 5.6. Oil Red O (ORO) Staining and Cell Steatosis Quantification

HCC cells (1 × 10^5^) were grown on coverslips in 6 well plates for 18h and exposed to either vehicle or different cyanotoxins (50 nM) as indicated. Cells were further treated with 100 µM oleic acid (Sigma) for 30 h. Cells were fixed with paraformaldehyde (4%) for 15 min and after washing with PBS, an Oil Red O (ORO) staining was performed as reported previously [74]. ORO-stained cells were observed under a Nikon Y-IDP microscope and images were captured. ORO staining-based cellular steatosis quantification was performed as described previously [75]. In brief, HepG2 and SK-Hep1 cells (1 × 10^4^/well) were grown in 6-well plates in triplicate and treated with cyanotoxins and oleic acid for 30 h. After ORO staining, cells were lysed and lysates (100 µL) were transferred to a 96-well plate, and the plates were read at 405 nm as described previously [74]. All experiments were repeated two to three times.

### 5.7. RNA-Seq and Analysis

Human hepatocytes, HepaRG, and HCC cells were exposed to vehicles or 50 nM MC-LR and CYN separately in triplicates for 72 h. The total RNA was isolated using TRIZOL reagents (Sigma; St. Louis, MO). RNA samples were quantified using Nanodrop One (ThermoScientific; Waltham, MA USA) and samples were sent to Novogene for RNA-seq analysis as per the company’s instructions. The sequencing libraries were prepared using the NEBNext Ultra RNA Library Prep Kit for Illumina following the manufacturer’s instructions (NEB, Ipswich, MA, USA). Genes with adjusted *p*-values < 0.05 and absolute log2 fold changes > 1 were called differentially expressed genes for each comparison. Heat maps and volcano plots were generated and gene ontology analysis was performed on the statistically significant set of genes by using GeneSCF software (v1.1). The human Gene Ontology list was used to cluster the set of genes based on their biological process/pathways. The RNA-seq array data were submitted to Annotare with accession number E-MTAB-12881 (https://www.ebi.ac.uk/fg/annotare/help/index.html, accessed on 13 March 2023). We also validated RNA-seq gene expression data after cyanotoxin exposure by RT/qPCR.

### 5.8. Statistical Analysis

The results are from independent duplicate or triplicate experiments and presented as means ± SEM. Differences between groups were analyzed using a two-tailed Student’s *t*-test. Statistical significance between means was determined by Graph Pad Prism 9 software (GraphPad Software Inc., La Jolla, CA, USA). Immunoblots differences among multiple means were assessed by one-way ANOVA and Bonferroni post hoc tests with Sigmaplot 14.5 Software (Palo Alto, CA, USA). A *p*-value of <0.05 was considered statistically significant.

## Figures and Tables

**Figure 1 toxins-15-00411-f001:**
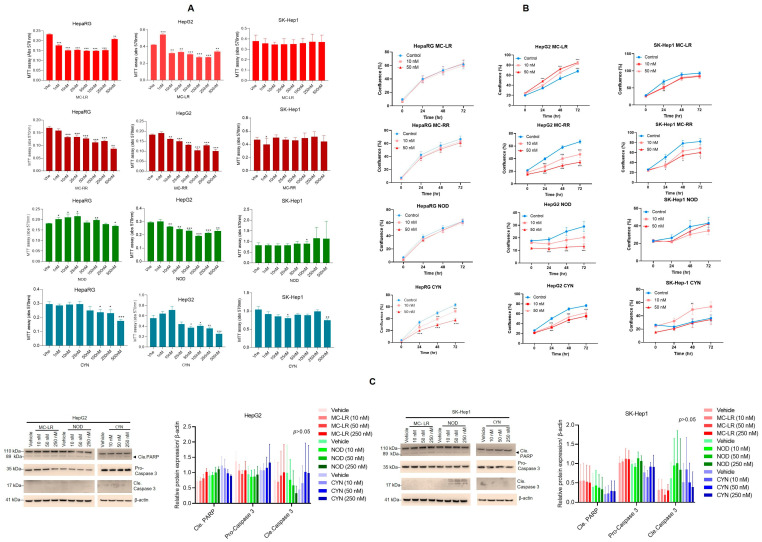
Effect of cyanotoxins on HepaRG, HepG2, and SK-Hep1 cell survival. (**A**) HepaRG, HepG2, and SK-Hep1 cells were exposed to 1 to 500 nM of MC-LR, MC-RR, NOD, and CYN for 72 h, and cell survival was analyzed by MTT assay (n = 6–9). * *p* < 0.05, ** *p* < 0.01, *** *p* < 0.001 compared with vehicle-treated cells. (**B**) HepaRG and HCC HepG2 and SK-Hep1 cells (5,000/well) were treated with 10 and 50 nM concentrations of MC-LR, MC-RR, NOD, and CYN for 72 h (n = 6), and the effect of cyanotoxins on real-time cell proliferation was analyzed by Incucyte. * *p* < 0.05, ** *p* < 0.01, *** *p* < 0.001 compared with vehicle-treated cells. Veh.-Vehicle. (**C**) HepG2 and SK-Hep1 cells were exposed to 10–250 nM of MC-LR, NOD, and CYN for 72 h (n = 3), and lysates were Western blotted with indicated antibodies. Band intensities were quantified by Image J (https://imagej.nih.gov/ij/; Version 1.53t, accessed on 25 May 2023) and are presented in the right-hand panels.

**Figure 2 toxins-15-00411-f002:**
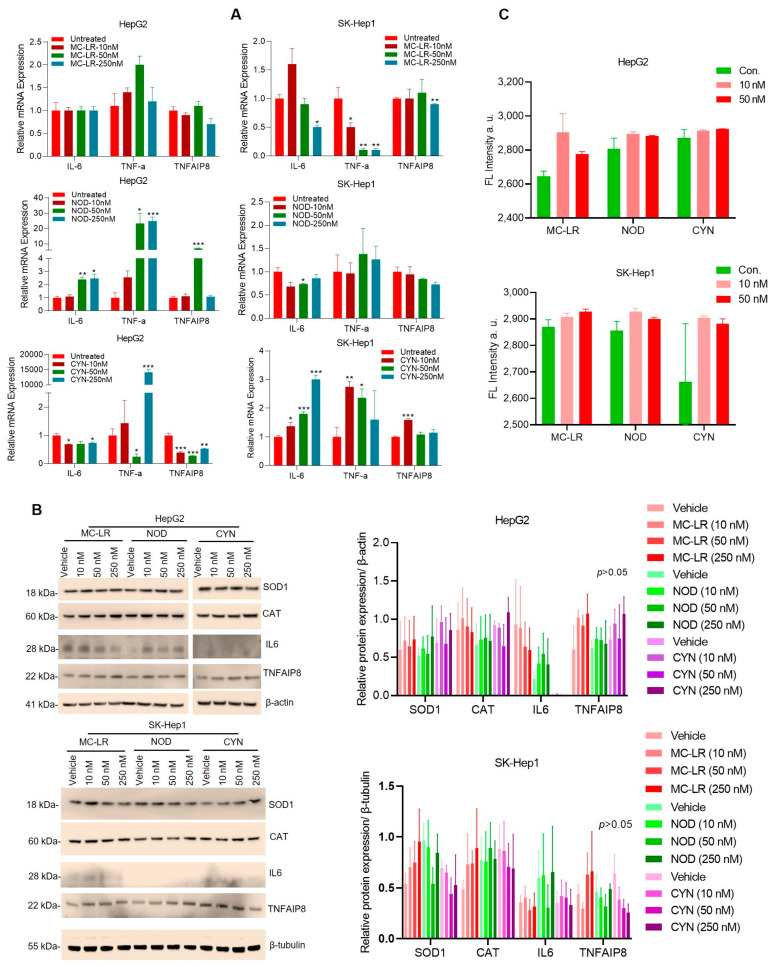
Cyanotoxins modulate inflammatory signaling in HCC cells. (**A**) HCC HepG2 and SK-Hep1 cells were exposed to MC-LR, NOD, and CYN for 72 h, and the expression of inflammatory genes was analyzed by RT-qPCR. * *p* < 0.05, ** *p* < 0.01, *** *p* < 0.001 compared with control cells. (**B)** HepG2 and SK-Hep1 cells were exposed to 10–250 nM of MC-LR, NOD, and CYN for 72 h (n = 3), and cell lysates (60 μg) were Western blotted with SOD1, CAT, IL6, TNFAIP8, β-actin, and β-tubulin antibodies. Band intensities were quantified by Image J (https://imagej.nih.gov/ij/; Version 1.53t, accessed on 25 May 2023) and are presented in the right-hand panels. (**C**) HepG2 and SK-Hep1 cells were grown in 96 well plates for 18 h (in triplicates) and cells were exposed to 10 and 50 nM of MC-LR, NOD, and CYN for 72 h. The endogenous green fluorescence (ROS-related) was quantified using FLUOstar^®^ Omega plate reader with excitation/emission at 485/520 nm and plotted.

**Figure 3 toxins-15-00411-f003:**
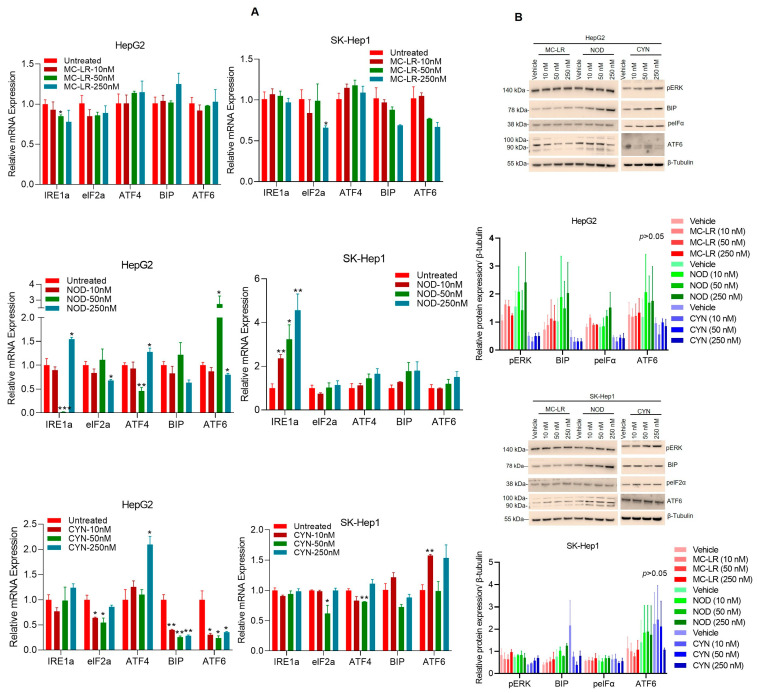
Cyanotoxins regulate UPR signaling in HCC cells. (**A**) HCC HepG2 and SK-Hep1 cells were exposed to MC-LR, NOD, and CYN for 72 h, as indicated, and the expression of UPR gene biomarkers was analyzed by RT/qPCR. * *p* < 0.05, ** *p* < 0.01, *** *p* < 0.001 related with untreated/vehicle-treated cells. (**B**) HepG2 and SK-Hep1 cells were exposed to 10–250 nM of MC-LR, NOD, and CYN for 72 h (n = 3), and cell lysates (60 μg) were Western blotted with pERK, BIP, peIF2α, ATF6, and β-tubulin. Western blot band intensities were quantified by Image J (https://imagej.nih.gov/ij/; Version 1.53t, accessed on 25 May 2023), as shown in the below panels.

**Figure 4 toxins-15-00411-f004:**
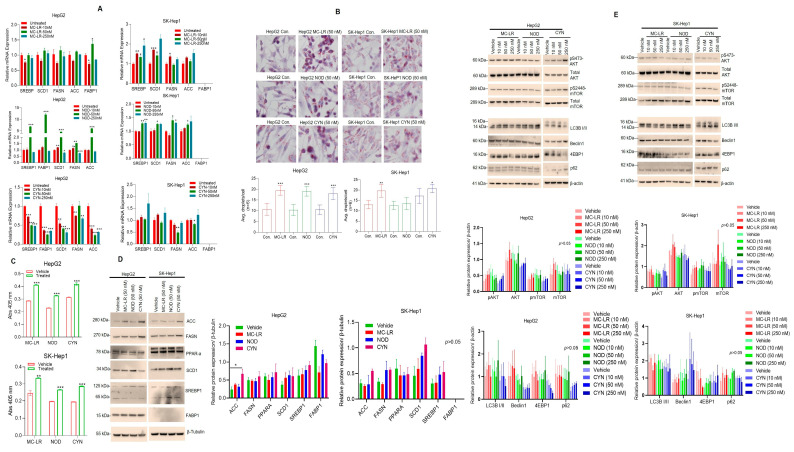
Cyanotoxins induce cell steatosis in HCC cells. (**A**) HepG2 and SK-Hep1 cells were incubated with 10, 50, and 250 nM of MC-LR, NOD, and CYN for 72 h, and lipogenic genes expression was analyzed by RT/qPCR. * *p* < 0.05, ** *p* < 0.01, *** *p* < 0.001 related with untreated cells. (**B**) HepG2 and SK-Hep1 cells were grown on coverslips in six-well plates and exposed to 50 nM of MC-LR, NOD, and CYN for 72 h. Cells were further exposed to 100 μM oleic acid (OA) in the presence of cyanotoxins for an additional 48 h and Oil Red O staining performed, and images were captured (upper panel) and the number of lipid droplets present in each cell (n = 6) was quantified manually and plotted (lower panels). * *p* < 0.05, ** *p* < 0.01, *** *p* < 0.001 related with OA treated control cells. (**C**) HepG2 and SK-Hep1 cells (1 × 10^5^/well) were cultured in 6-well plates in triplicates and exposed to 50 nM of MC-LR, NOD, and CYN for 24 h, and then treated with 100 μM oleic acid for additional 48 h; after ORO staining, cells were lysed in 100 μL of 1 × cell lysis buffer and the absorbance at 405 nm was measured. ** *p* < 0.01, *** *p* < 0.001 related to the vehicle and OA-treated cells. (**D**) HepG2 and SK-Hep1 cells were exposed to 50 nm of MC-LR, NOD, and CYN for 72 h (n = 3), and expression of lipogenic proteins was analyzed by immunoblotting as indicated. Band intensities were quantified by Image J (https://imagej.nih.gov/ij/; Version 1.53t, accessed on 25 May 2023) and are presented. * *p* < 0.05. (**E**) HepG2 and SK-Hep1 cells (n = 3) were exposed to indicated cyanotoxins (10, 50, and 250 nM) and the expression of AKT/mTOR and autophagy markers were analyzed by Western blotting. Band intensities were quantified by Image J (https://imagej.nih.gov/ij/; Version 1.53t, accessed on 25 May 2023) and are presented (below panels).

**Figure 5 toxins-15-00411-f005:**
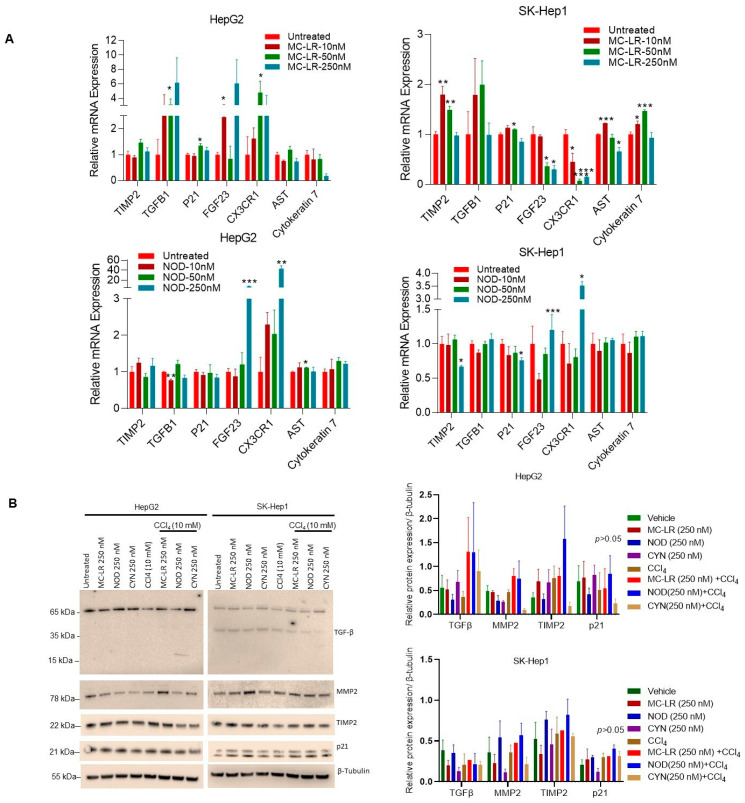
Cyanotoxins modulate cell fibrosis in HCC cells. (**A**) HepG2 and SK-Hep1 cells were exposed to 10, 50, and 250 nM of MC-LR, NOD, and CYN for 72 h and the expression of fibrogenic genes was analyzed by RT/qPCR (left and right panels). * *p* < 0.05, ** *p* < 0.01, *** *p* < 0.001 related with control cells. (**B**) The effect of cyanotoxins and cyanotoxins plus CCl_4_ on the expression of fibrogenic protein markers was analyzed (n = 3) by Western blotting. Western blots band intensities were quantified by Image J (https://imagej.nih.gov/ij/; Version 1.53t, accessed on 25 May 2023) and are presented (right panels).

**Figure 6 toxins-15-00411-f006:**
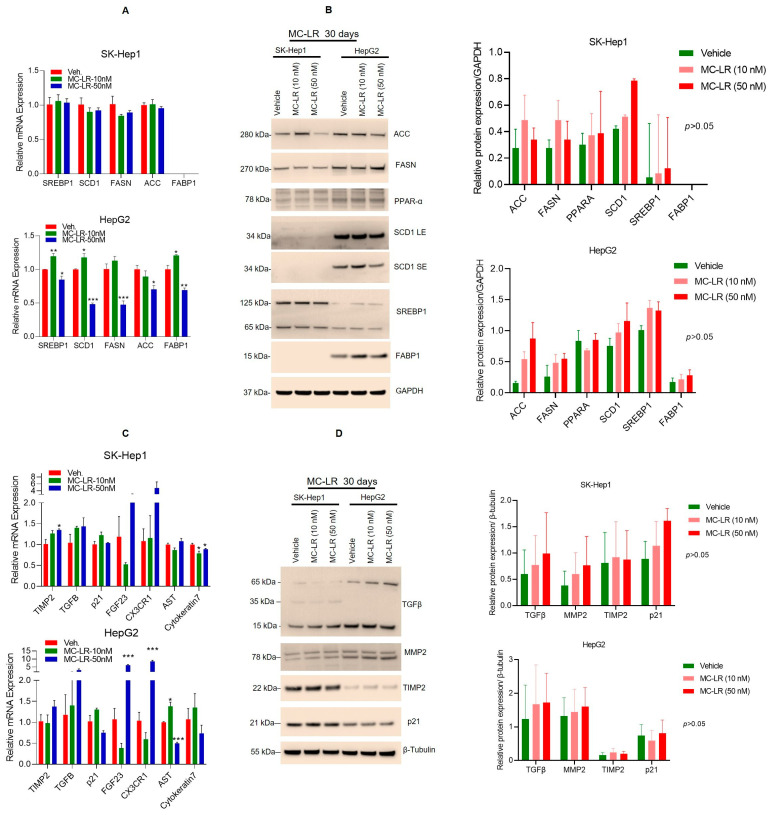
Chronic exposure to cyanotoxins induced lipogenic and fibrosis biomarkers in HCC cells. (**A**–**D**) SK-Hep1 and HepG2 cells were exposed to MC-LR (10 and 50 nM) for 30 days and the genes and proteins expression of lipogenic (**A**,**B**) and fibrogenic markers and protein expression (**C**,**D**) were analyzed by RT/qPCR and immunoblotting, respectively (n = 3). LE—long exposure; SE—short exposure. Immunoblot band intensities were quantified by Image J (https://imagej.nih.gov/ij/; Version 1.53t, accessed on 25 May 2023). * *p* < 0.05, ** *p* < 0.01, *** *p* < 0.001 compared with vehicle-treated cells.

**Figure 7 toxins-15-00411-f007:**
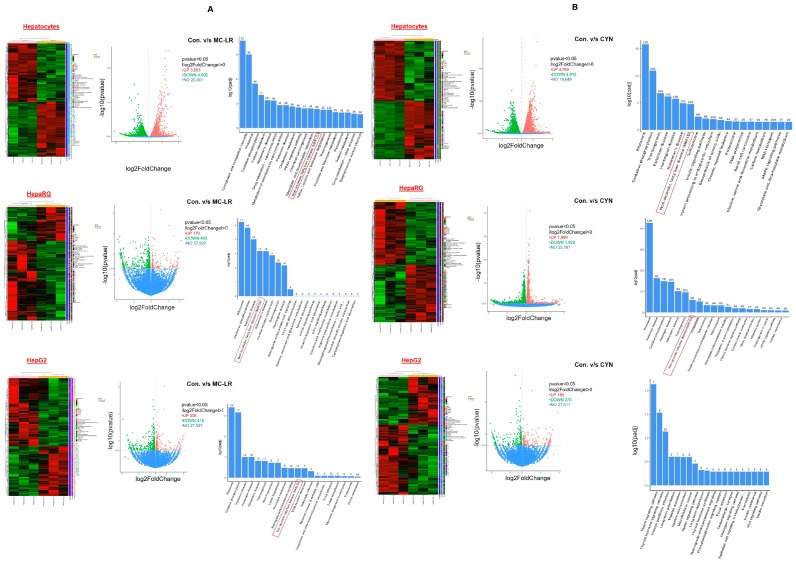
Effect of MC-LR and CYN on global gene expression. RNA-Seq assays. (**A**) Left panels: Heat map represents the impact of MC-LR (50 nM) for 72 h (*n* = 3) on global gene expression in human hepatocytes, HepaRG, and HepG2 cells. Middle panels: Volcanic plot represents differential gene expression levels between vehicle-treated and MC-LR-treated in human hepatocytes, HepaRG, and HepG2 cells. Right panels: The gene ontology (GO) enrichment analysis of the differentially expressed mRNAs in human hepatocytes, HepaRG, and HepG2 cells after MC-LR exposure for 72 h. The impact of MC-LR on the modulation of NAFLD pathways is presented (red box). (**B**) RNA-Seq assays. Left panels: Heat map represents the impact of CYN (50 nM) for 72h (*n* = 3 samples) on global gene expression in human hepatocytes, HepaRG, and HepG2 cells. Middle panels: Volcanic plot represents differential gene expression levels between vehicle-treated and CYN-treated (50 nM) in human hepatocytes, HepaRG, and HepG2 cells. Right panels: The gene ontology (GO) enrichment analysis of the differentially expressed mRNAs in human hepatocytes, HepaRG, and HepG2 cells after CYN exposure for 72 h. The impact of CYN on the modulation of NAFLD pathways is presented (red box).

**Figure 8 toxins-15-00411-f008:**
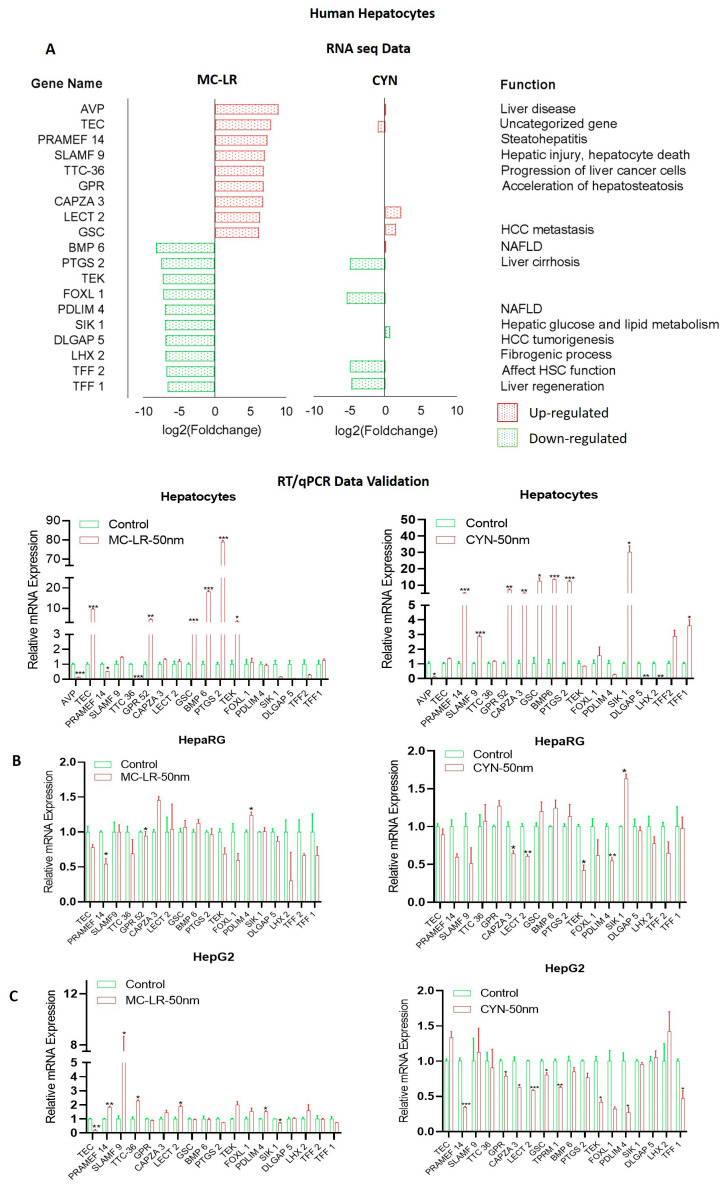
Cyanotoxins modulate NAFLD-related gene expression in liver cell models. (**A**) The effect of MC-LR and CYN (50 nM) for 72 h exposure on the regulation of NAFLD and fatty liver diseases-related gene expression in hepatocytes was analyzed using RNA−Seq data (upper panel). The function of these key genes in the modulation of liver diseases is presented after a literature search. Validation of RNA−Seq data was performed by RT−qPCR using human hepatocyte RNA samples (lower panels). * *p* < 0.05, ** *p* < 0.01, *** *p* < 0.001 compared with control cells. (**B**,**C**) Validation of RNA−Seq data was performed by RT−qPCR using HepaRG and HepG2 RNA samples after treatment of cells with 50 nM of MC-LR and CYN for 72 h. * *p* < 0.05, ** *p* < 0.01, *** *p* < 0.001 compared with vehicle-treated cells.

## Data Availability

The published article includes all data sets generated/analyzed for this study.

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
