# Peer review of "Cyanotoxins Increase Cytotoxicity and Promote Nonalcoholic Fatty Liver Disease Progression by Enhancing Cell Steatosis"

_toxins, 2023, doi:10.3390/toxins15070411_

Round 1

Reviewer 1 Report

Journal: Toxins (ISSN 2072-6651)

Manuscript ID: toxins-2372558

Type: Article

Title: Microcystins increase cytotoxicity and promote non-alcoholic fatty liver disease progression by enhancing cell steatosis

This manuscript aimed to investigate the effects and impact of low concentrations of MC-LR, MC-RR, NOD, and CYN on hepatotoxicity, hepatic steatosis, and fibrosis in hepatocytes, HepaRG, and HCC cells. The results are interesting. However, the presentation and writing are not good. Please note that NOD and CYN are not MCs. I have the following comments and suggestions for the authors to improve the quality of manuscript.

1. Abstract

Lines 13-15

“Exposure to MCs regulates lipogenic gene expression and induces cell steatosis and fibrosis signaling in HCC cells. MC-LR, NOD, and CYN activate AKT/mTOR, increase p62 expression, and inhibit autophagy.”

Please change “regulates”, “induces”, “activate”, “increase” and “inhibit” to “regulated”, “induced”, “activated”, “increased” and “inhibited”, respectively. There are a number of grammatical errors and instances of badly worded/constructed sentences. Please check the manuscript and refine the language carefully. Please ask an English native speaker to correct the language. There are many companies which can offer this service.

2. Line 20

Please change “promotes” to “promote”.

3. Section “1. Introduction”

Paragraphs 1-4, lines 26-83

This manuscript studied toxicity of microcystin -LR (MC-LR), microcystin-RR (MC-RR), nodularin (NOD), and cylindrospermopsin (CYN). But in the introduction, MC-RR, NOD and CYN were not mentioned. Please insert some text about occurrence and toxicity of MC-RR, NOD, and CYN. Please note that NOD and CYN are not MCs. There are many expressions are wrong in the manuscript. The authors had considered that NOD and CYN are not MCs.

4. Lines 55-56

“In the United States, microcystin-LR (MC-LR) is the most studied and considered a significant algal bloom toxin [8].”

Microcystin-LR (MC-LR) is the most studied not only in the United States, but also all over the world. Please delete “In the United States”.

5. Lines 63-64

“Although the liver is a primary target for microcystin, different organs can be impacted such as the kidneys and the gut [11-13].”

Brain, spleen and gonad can also be impacted. Please read and cite the following paper.

Chen et al., 2021. Challenges of using blooms of Microcystis spp. in animal feeds: A comprehensive review of nutritional, toxicological and microbial health evaluation. https://doi.org/10.1016/j.scitotenv.2020.142319

Abdallah et al., 2021. Cyanotoxins and Food Contamination in Developing Countries: Review of Their Types, Toxicity, Analysis, Occurrence and Mitigation Strategies. https://doi.org/10.3390/toxins13110786

6. Section “2. Results”

Lines 90-91

“Although about 279 different MCs have been identified so far, in the current study, we selected four MCs: MC-LR, MC-RR, NOD, and CYN.”

NOD and CYN are not MCs.

7. Lines 92-96

“To understand the impact of low concentrations of MCs on liver cell metabolism and survival, we exposed terminally differentiated human bipotent progenitor liver cells HepaRG and two HCC cell lines, HepG2 and SK-Hep1 cells to increasing concentrations (1 to 500 nM) of MC-LR, MC-RR, NOD, and CYN for 72h and cell metabolic activities were analyzed by MTT assay (Fig. 1A).”

What do you mean by “terminally differentiated” for human bipotent progenitor liver cells HepaRG? HepaRG cells are capable of differentiating into two different cell phenotypes (i.e., biliary-like and hepatocyte-like cells). Please insert more details in the revised manuscript.

8. Fig. 1A

For HepaRG cells, why the effects were not the largest, at the largest concentration of MC-LR, 500 nM MC-LR? It did not show dose-dependent effects. Please insert some explanations in the revised manuscript.

9. Fig. 1B

Please add experiments and results of HepaRG cells.

10. Fig. 1AB

In Fig.1A, MC-LR decreased cell viability in HepG2 cells. However, in Fig.1B, MC-LR increased cell proliferation in HepG2 cells. Please insert some explanations in the revised manuscript.

11. Fig. 1AB

In Fig.1A, CYN decreased cell viability in SK-Hep1 cells. However, in Fig.1B, CYN increased cell proliferation in SK-Hep1 cells. Please insert some explanations in the revised manuscript.

12. Fig. 1C

Did you perform repeated experiments? Please add quantitative and statistical results of immunoblotting.

13. Fig. 2B

Did you perform repeated experiments? Please add quantitative and statistical results of immunoblotting.

14. Fig. 3B

Did you perform repeated experiments? Please add quantitative and statistical results of immunoblotting.

15. Supplementary Fig. S3A

Did you perform repeated experiments? Please add quantitative and statistical results of cell cycle, percentages of cells at different phases.

16. Fig. 4B

Did you perform repeated experiments? Please add quantitative and statistical results of immunoblotting.

17. Supplementary Fig. S4

Did you perform repeated experiments? Please add quantitative and statistical results of immunoblotting.

18. Fig. 4D

In the figure, the absorbance was 450 nm. However, in the figure captions, the absorbance was 405 nm. Please check it.

19. Fig. 4E

Did you perform repeated experiments? Please add quantitative and statistical results of immunoblotting.

20. Fig. 5B

Did you perform repeated experiments? Please add quantitative and statistical results of immunoblotting.

21. Figure 6A

Please insert results of FABP1 in SK-Hep1 cells.

22. Fig. 6B

Did you perform repeated experiments? Please add quantitative and statistical results of immunoblotting.

22. Fig. 6D

Did you perform repeated experiments? Please add quantitative and statistical results of immunoblotting.

23. Lines 376-380

“Interestingly, MC-LR or CYN upregulates the NAFLD pathway in hepatocytes (Fig. 6A &B, upper panels). In HepaRG cells, MC-LR or CYN upregulates oxidative phosphorylation, ribosomal, Parkinson's disease, Alzheimer's disease, mTOR as well as NAFLD pathway (Fig. 7A &B, middle panels)”

The results of GeneRatio/gene ontology analysis were not presented in Figure 6 or 7.

24. Lines 385-395

“Importantly, RNA-Seq data suggest that MC-LR and CYN exposure shows upregulation of the NAFLD pathway in hepatocytes, HepaRG, and HepG2 cells suggesting that MCs modulate NAFLD progression (Fig. 7A & B). Further, we analyzed the RNA-seq data and identified key genes that contribute to NAFLD progression directly or indirectly. RNA-seq data revealed that MC-LR exposure upregulates AVP, TEC, PRAMEF14, SLAMF9, TTC-36, GPR5, CAPZA3, LECT2, and GSC genes in hepatocytes (Fig. 8A). Whereas, MC-LR exposure downregulates BMP6, PTGS2, TEK, FOXL1, PDLIM4, SIK1, DLAGP5, LHX2, TFF2, and TFF1 genes. (Fig. 8A). CYN exposure to hepatocytes shows only LECT2 and GCS gene upregulation and PTGS2, FOXL1, TFF2, and TFF1 downregulation (Fig. 8A). We performed the literature search, and the roles of these genes in liver diseases were presented (Fig. 8A).”

The results of RNA-Seq were not presented in Figure 7 or 8.

25. Figure 8B, C, D

“Validation of key genes related to NAFLD and fatty liver diseases from RNA-Seq data was carried out by RT-qPCR”

Which results are shown? RNA-Seq or RT-qPCR? Please refer to Figure 4 of the paper by Zhang et al. (2020) and re-draw your figures. Then readers can understand the data of both RNA-Seq and RT-qPCR from a same figure.

Zhang et al. 2020. Microcystin-LR-induced changes of hepatopancreatic transcriptome, intestinal microbiota, and histopathology of freshwater crayfish (Procambarus clarkii). https://doi.org/10.1016/j.scitotenv.2019.134549

ok

Author Response

Comments and Suggestions for Authors

Journal: Toxins (ISSN 2072-6651)

Manuscript ID: toxins-2372558

Type: Article

Title: Microcystins increase cytotoxicity and promote non-alcoholic fatty liver disease progression by enhancing cell steatosis

Reviewer 1

This manuscript aimed to investigate the effects and impact of low concentrations of MC-LR, MC-RR, NOD, and CYN on hepatotoxicity, hepatic steatosis, and fibrosis in hepatocytes, HepaRG, and HCC cells. The results are interesting. However, the presentation and writing are not good. Please note that NOD and CYN are not MCs. I have the following comments and suggestions for the authors to improve the quality of manuscript.

 Response: Thanks for your comments. We modified the whole text and grammatical errors are corrected in the revised version.

  1. Abstract

Lines 13-15

“Exposure to MCs regulates lipogenic gene expression and induces cell steatosis and fibrosis signaling in HCC cells. MC-LR, NOD, and CYN activate AKT/mTOR, increase p62 expression, and inhibit autophagy.”

Please change “regulates”, “induces”, “activate”, “increase” and “inhibit” to “regulated”, “induced”, “activated”, “increased” and “inhibited”, respectively. There are a number of grammatical errors and instances of badly worded/constructed sentences. Please check the manuscript and refine the language carefully. Please ask an English native speaker to correct the language. There are many companies which can offer this service.

Response:  Changes made as suggested.

  1. Line 20

Please change “promotes” to “promote”.

Response: Done.

  1. Section “1. Introduction”

Paragraphs 1-4, lines 26-83

This manuscript studied toxicity of microcystin -LR (MC-LR), microcystin-RR (MC-RR), nodularin (NOD), and cylindrospermopsin (CYN). But in the introduction, MC-RR, NOD and CYN were not mentioned. Please insert some text about occurrence and toxicity of MC-RR, NOD, and CYN. Please note that NOD and CYN are not MCs. There are many expressions are wrong in the manuscript. The authors had considered that NOD and CYN are not MCs.

Response: We incorporated the studies related to cyanotoxins MC-LR, MC-RR, NOD, and CYN toxicity and related liver biology in the introduction section. We also addressed that MC-LR, MC-RR, NOD, and CYN are cyanotoxins and updated related literature in the revised version.

  1. Lines 55-56

“In the United States, microcystin-LR (MC-LR) is the most studied and considered a significant algal bloom toxin [8].”

Microcystin-LR (MC-LR) is the most studied not only in the United States, but also all over the world. Please delete “In the United States”.

Response: Modified as suggested. 

  1. Lines 63-64

“Although the liver is a primary target for microcystin, different organs can be impacted such as the kidneys and the gut [11-13].”

Brain, spleen and gonad can also be impacted. Please read and cite the following paper.

Chen et al., 2021. Challenges of using blooms of Microcystis spp. in animal feeds: A comprehensive review of nutritional, toxicological and microbial health evaluation. https://doi.org/10.1016/j.scitotenv.2020.142319

Abdallah et al., 2021. Cyanotoxins and Food Contamination in Developing Countries: Review of Their Types, Toxicity, Analysis, Occurrence and Mitigation Strategies. https://doi.org/10.3390/toxins13110786

Response: These studies and other related information are added in the introduction section.

  1. Section “2. Results”

Lines 90-91

“Although about 279 different MCs have been identified so far, in the current study, we selected four MCs: MC-LR, MC-RR, NOD, and CYN.”

NOD and CYN are not MCs.

Response: Modified text in the revised version of the manuscript.

  1. Lines 92-96

“To understand the impact of low concentrations of MCs on liver cell metabolism and survival, we exposed terminally differentiated human bipotent progenitor liver cells HepaRG and two HCC cell lines, HepG2 and SK-Hep1 cells to increasing concentrations (1 to 500 nM) of MC-LR, MC-RR, NOD, and CYN for 72h and cell metabolic activities were analyzed by MTT assay (Fig. 1A).”

What do you mean by “terminally differentiated” for human bipotent progenitor liver cells HepaRG? HepaRG cells are capable of differentiating into two different cell phenotypes (i.e., biliary-like and hepatocyte-like cells). Please insert more details in the revised manuscript.

Response: The HepaRG™ cell line is an immortalized hepatic cell line that retains many characteristics of primary human hepatocytes. According to manufacturers' and suppliers’ suggestions, HepaRG™ cells are terminally differentiated and provided in a convenient cryopreserved format and suitable for drug metabolism studies. Related references and supplier information are cited in the text.

  1. Fig. 1A

For HepaRG cells, why the effects were not the largest, at the largest concentration of MC-LR, 500 nM MC-LR? It did not show dose-dependent effects. Please insert some explanations in the revised manuscript.

Response: A probable hypothesis, low doses have a dose-dependent effect but since HepaRG is immortalized, high doses could have triggered survival mechanisms to overcome the acute insult. Explanation added in the text.  

  1. Fig. 1B

Please add experiments and results of HepaRG cells.

Response: Done and inserted in Figure 1B.

  1. Fig. 1AB

In Fig.1A, MC-LR decreased cell viability in HepG2 cells. However, in Fig.1B, MC-LR increased cell proliferation in HepG2 cells. Please insert some explanations in the revised manuscript.

Response: Explanation added in the text.  Although cell metabolic activities and cell proliferation and associated with each other but exposure to MC-LR may behave differently.  HepG2 cells are carcinogenic and immortalized, different doses could have triggered dysregulated cell survival mechanisms differentially to overcome the acute insult 

  1. Fig. 1AB

In Fig.1A, CYN decreased cell viability in SK-Hep1 cells. However, in Fig.1B, CYN increased cell proliferation in SK-Hep1 cells. Please insert some explanations in the revised manuscript.

Response: Added.

12 and 14-17. Immunoblotting: Did you perform repeated experiments? Please add quantitative and statistical results of immunoblotting.

Response:  Immunoblotting was performed two to three times, however, one set of immunoblotting data presented in the manuscript where the endogenous control expression remains constant.  We also quantify the relative band intensities presented below in Fig. 1C, Fig. 3B, and Fig. 6D.

  1. Did you perform repeated experiments? Please add quantitative and statistical results of cell cycle, percentages of cells at different phases.

Response: We decided to remove Fig. S3 and Fig. S4 data from this publication for another publication.

  1. Fig. 4D

In the figure, the absorbance was 450 nm. However, in the figure captions, the absorbance was 405 nm. Please check it.

Response:  Corrected. 

  1. Figure 6A

Please insert results of FABP1 in SK-Hep1 cells.

Response: The expression of FABP1 in SK-HEP1 was not detected under our experimental conditions. 

  1. Lines 376-380

“Interestingly, MC-LR or CYN upregulates the NAFLD pathway in hepatocytes (Fig. 6A &B, upper panels). In HepaRG cells, MC-LR or CYN upregulates oxidative phosphorylation, ribosomal, Parkinson's disease, Alzheimer's disease, mTOR as well as NAFLD pathway (Fig. 7A &B, middle panels)”

The results of GeneRatio/gene ontology analysis were not presented in Figure 6 or 7.

Response: Please see Figures 7A and B right panels. GeneRatio/gene ontology analysis and the impact of cyanotoxin(s) on cellular pathways were presented.

  1. Lines 385-395

“Importantly, RNA-Seq data suggest that MC-LR and CYN exposure shows upregulation of the NAFLD pathway in hepatocytes, HepaRG, and HepG2 cells suggesting that MCs modulate NAFLD progression (Fig. 7A & B). Further, we analyzed the RNA-seq data and identified key genes that contribute to NAFLD progression directly or indirectly. RNA-seq data revealed that MC-LR exposure upregulates AVP, TEC, PRAMEF14, SLAMF9, TTC-36, GPR5, CAPZA3, LECT2, and GSC genes in hepatocytes (Fig. 8A). Whereas, MC-LR exposure downregulates BMP6, PTGS2, TEK, FOXL1, PDLIM4, SIK1, DLAGP5, LHX2, TFF2, and TFF1 genes. (Fig. 8A). CYN exposure to hepatocytes shows only LECT2 and GCS gene upregulation and PTGS2, FOXL1, TFF2, and TFF1 downregulation (Fig. 8A). We performed the literature search, and the roles of these genes in liver diseases were presented (Fig. 8A).”

The results of RNA-Seq were not presented in Figure 7 or 8.

Response: Thank you for the comment.  The role and biological significance of cyanotoxin-related gene expression were included in the discussion section after the literature search.

  1. Figure 8B, C, D

“Validation of key genes related to NAFLD and fatty liver diseases from RNA-Seq data was carried out by RT-qPCR”

Which results are shown? RNA-Seq or RT-qPCR?

Please refer to Figure 4 of the paper by Zhang et al. (2020) and re-draw your figures. Then readers can understand the data of both RNA-Seq and RT-qPCR from a same figure.

Response: Fig. 8A summarized differential gene expression from RNA seq data after cyanotoxins exposure and their role in liver diseases using human hepatocytes.    Fig. 8B is a Validation of RNA seq data using human hepatocytes.

In Fig, 8C and D we analyzed the expression of genes that are differentially expressed in human hepatocytes using HCC HepG2 cells after cyanotoxins exposure.  We agree that we should replot the figure, however, we clearly stated in the text and labeled the figure clearly in the revised version.   

Reviewer 2 Report

Manuscript Number: TOXINS-2372558

The manuscript entitled “Microcystins increase cytotoxicity and promote non-alcoholic fatty liver disease progression by enhancing cell steatosis” is an original experimental study which provides results about the hepatotoxicity induced by some cyanotoxins in different cell models. The study presents a large number of results and responds to a lot of laboratory work. The topic is important and is part of the journal's scope. However, the fact that MCs are hepatotoxic is widely known. It is therefore necessary for the authors to highlight and justify the novelty of these studies.

In addition, the manuscript presents an underlying problem as it considers all 4 toxins as MCs. Of the 4 cyanotoxins studied, only MC-LR and MC-RR are MCs. NOD and CYN are cyanotoxins with a different structure, physico-chemical properties etc. than MCs. Therefore, the manuscript should have independently presented and discussed the results obtained for each of these toxins. For example, in the introductory section it only focuses on MC-LR (which is the best known) but does not present data on the rest of the cyanotoxins investigated.

Moreover, published data similar to those presented in this manuscript already exist, such as cytotoxicity in HepG2, ROS etc. for some of these cyanotoxins.

I therefore suggest major changes (major revision) for publication in the TOXINS journal.

In general, and for the improvement of this manuscript, the authors should take into account:

-Complete the introduction section with data for all cyanotoxins, not only MC-LR.

-MC-LR is classified in group 2B by IARC. This information is important and should be indicated in the text.

-There is a lack of data on the concentrations of these toxins found in the environment.

-Very old data are presented 2009-2010. Please, these data could be updated.

-Please explicitly detail the objective of this study, what it contributes with respect to what has already been investigated (novelty).

-The results are difficult to follow in many of the figures. I recommend introduce tables with the most relevant results sorted by toxin/cell line/parameters investigated.

-In general, the figure captions are too long. Please summarize it.

-The discussion section could be improved with more in vitro data and a focus on each toxin.

-The conclusion should be better exposed. It is too short.

Author Response

Reviewer 2

Manuscript Number: TOXINS-2372558

The manuscript entitled “Microcystins increase cytotoxicity and promote non-alcoholic fatty liver disease progression by enhancing cell steatosis” is an original experimental study which provides results about the hepatotoxicity induced by some cyanotoxins in different cell models. The study presents a large number of results and responds to a lot of laboratory work. The topic is important and is part of the journal's scope. However, the fact that MCs are hepatotoxic is widely known. It is therefore necessary for the authors to highlight and justify the novelty of these studies.

Response: Thanks for your comments. The important role and novelty of cyanotoxins in NAFLD progression are now mentioned in the revised manuscript.

In addition, the manuscript presents an underlying problem as it considers all 4 toxins as MCs. Of the 4 cyanotoxins studied, only MC-LR and MC-RR are MCs. NOD and CYN are cyanotoxins with a different structure, physico-chemical properties etc. than MCs. Therefore, the manuscript should have independently presented and discussed the results obtained for each of these toxins. For example, in the introductory section it only focuses on MC-LR (which is the best known) but does not present data on the rest of the cyanotoxins investigated.

Response: We incorporated the studies related to MC-LR, MC-RR, NOD, and CYN toxicity and related liver biology.

Moreover, published data similar to those presented in this manuscript already exist, such as cytotoxicity in HepG2, ROS etc. for some of these cyanotoxins.

Response: We agreed with the reviewer's comments but here we analyzed the impact of 3-4 cyanotoxins and four cell lines together and the results are clearly stated in the revised manuscript.  

I therefore suggest major changes (major revision) for publication in the TOXINS journal.

Response: We made substantial changes as suggested by all reviewers and editors. 

In general, and for the improvement of this manuscript, the authors should take into account:

-Complete the introduction section with data for all cyanotoxins, not only MC-LR.

Response: Done please see revised version.

-MC-LR is classified in group 2B by IARC. This information is important and should be indicated in the text.

Response: Statement added. “According to the International Agency for Research on Cancer (IARC 2010), MC-LR is classified as a Group 2B possible carcinogen, if humans are exposed to MC-LR chronically”.

-There is a lack of data on the concentrations of these toxins found in the environment.

Response: Information added, please see the revised version.

-Very old data are presented 2009-2010. Please, these data could be updated.

Response:  Recent information added. Please see the revised version.

-Please explicitly detail the objective of this study, and what it contributes with respect to what has already been investigated (novelty).

Response: Presented in the revised manuscript.

-The results are difficult to follow in many of the figures. I recommend introduce tables with the most relevant results sorted by toxin/cell line/parameters investigated.

Response: We agreed with the reviewer’s comments, however, we used 3 to 4 cyanotoxins and 3-4 cell lines. The revised manuscript addresses all issues and is sorted out and we believe there is no need to add additional tables.  

-In general, the figure captions are too long. Please summarize it.

Response: We shortened figure legends.

-The discussion section could be improved with more in vitro data and a focus on each toxin.

Response: Done. A whole new paragraph was added in the discussion section.   

-The conclusion should be better exposed. It is too short.

Response: Conclusions modified in the revised manuscript.

Round 2

Reviewer 1 Report

Journal: Toxins (ISSN 2072-6651)

Manuscript ID: toxins-2372558-peer-review-v2

Type: Article

Title: Cyanotoxins increase cytotoxicity and promote nonalcoholic fatty liver disease progression by enhancing cell steatosis

This manuscript aimed to investigate the effects and impact of low concentrations of MC-LR, MC-RR, NOD, and CYN on hepatotoxicity, hepatic steatosis, and fibrosis in hepatocytes, HepaRG, and HCC cells. The results are interesting. However, the presentation and writing are not good. I have the following comments and suggestions for the authors to improve the quality of manuscript.

1. Section “Key Contribution”

Lines 23-24

“Exposure to cyanotoxins; microcystin-LR (MC-LR), microcystin-RR (MC-RR), nodularin (NOD), and cylindrospermopsin (CYN) increased liver cell toxicity.”

Please change “;” to “,”.

2. Section “1. Introduction”

Lines 38-39

“People may be exposed to cyanotoxins by drinking or swimming in contaminated water [1].”

People can also be exposed to cyanotoxins by eating contaminated aquatic products. Please cite the following paper. In fact, you have already cited this paper, the present ref. [20]

Challenges of using blooms of Microcystis spp. in animal feeds: A comprehensive review of nutritional, toxicological and microbial health evaluation. Sci Total Environ, 2021. 764: p. 142319.

3. Lines 68-70

“Cyanotoxins are divided into cyclic peptides, alkaloids, lipopeptides, non-protein amino acids, and lipoglycans and so far, 279 cyanotoxins have been identified [13].”

Please change to “Cyanotoxins are divided into cyclic peptides, alkaloids, lipopeptides, non-protein amino acids, and lipoglycans. So far, 279 microcystins (MCs) have been identified [13].”

4. Lines 80-88

“Although microcystins primarily target the liver, different organs can be impacted such as the gut, kidneys, brain, heart, lungs, and the reproductive system [17-19]. The genus Microcystis contains several species including M. aeruginosa are known to produce several microcystins (MCs) that enhanced toxicity in the kidney, intestine, spleen, liver, and other organs of several fish species [20], and a recent report suggests that cyanotoxins also affect several human organs such as brain, stomach, small and large intestine, lungs, kidneys, skin, and liver, also increased risk of several clinical symptoms [21]. Recent numerous studies indicated that cyanotoxin exposure is associated with hepatic inflammation, gastroenteritis, and liver cancers [18, 19, 22, 23].”

The text is repeated, about muti-organ toxicity of MCs. Please condense the sentences. You can keep the cited references, as they are about different species.

5. Lines 103-104

“Furthermore, MC-LR-induced nonalcoholic steatohepatitis (NASH) in rats when rats were fed a high-fat or high-cholesterol diet [38].”

Please change “MC-LR-induced” to “MC-LR induced”.

6. Section “2. Results”

Lines 128-129

“Although so far about 279 different cyanotoxins have been identified, in the current study we selected four cyanotoxins: MC-LR, MC-RR, NOD, and CYN.”

279 different MCs have been identified. Please delete this sentence.

7. Line 156

Please change “nm” to “nM”.

8. Lines 164-165

“In SK-Hep1 cells, MC-LR increased cleaved PARP expression (Fig. 1C, middle panel).”

Fig. 1C, the results show no increase of cleaved PARP expression in SK-Hep1 cells. Please check it.

9. Fig. 1C

Please add statistical results of immunoblotting from repeated experiments. Please refer to Figures 3, 5 and 6 from the following paper.

Regulation of heat shock protein 27 phosphorylation during microcystin-LR-induced cytoskeletal reorganization in a human liver cell line. Toxicology Letters 207 (2011) 270–277. doi:10.1016/j.toxlet.2011.09.025.

10. Lines 168-169

“low concentrations of cyanotoxins enhance cytotoxicity”

What do you mean?

11. Fig. 1B

Please re-analyze the data of HepG2-CYN.

12. Figures 1-8

Please check and unify capital and small letters of p. For example, p is small in Fig. 1, while is capital in Fig. 5.

13. Line 185

Please delete “and its”.

14. Fig. 2B

Please add quantitative and statistical results of immunoblotting from repeated experiments. Please refer to Figures 3, 5 and 6 from the following paper.

Regulation of heat shock protein 27 phosphorylation during microcystin-LR-induced cytoskeletal reorganization in a human liver cell line. Toxicology Letters 207 (2011) 270–277. doi:10.1016/j.toxlet.2011.09.025.

15. Lines 224-226

“As shown in Fig. 3B, MC-LR increased the expression of pERK, and BIP in HepG2 cells and pERK, BIP, peIF2a, and ATF6 in SK-Hep1 cells (Fig. 3B, upper and lower panels).”

Fig. 3B, the results show no increase of pERK expression in HepG2 or SK-Hep1 cells. Please check it.

16. Fig. 3B

Please add statistical results of immunoblotting from repeated experiments. Please refer to Figures 3, 5 and 6 from the following paper.

Regulation of heat shock protein 27 phosphorylation during microcystin-LR-induced cytoskeletal reorganization in a human liver cell line. Toxicology Letters 207 (2011) 270–277. doi:10.1016/j.toxlet.2011.09.025.

17. Line 245

Please change “modulates” to “modulate”.

18. Fig. 4A

Please add results of FABP1 in SK-Hep1 cells.

19. Fig. 4BE

Please add quantitative and statistical results of immunoblotting from repeated experiments. Please refer to Figures 3, 5 and 6 from the following paper.

Regulation of heat shock protein 27 phosphorylation during microcystin-LR-induced cytoskeletal reorganization in a human liver cell line. Toxicology Letters 207 (2011) 270–277. doi:10.1016/j.toxlet.2011.09.025.

20. Fig. 4B

Why are there 3 bands for SREBP1?

21. Lines 300-302

“Exposure of MC-LR significantly increased mRNA expression of TGFB1, p21 (~ 6 fold), cytokeratin 7 (~4.2 fold), and FGF-23 (~2.2 fold) in HepG2 cells compared with vehicle-treated cells.”

In Fig. 5A, the results show no increase of cytokeratin 7 (~4.2 fold) in HepG2 cells. Please check it.

22. Fig. 5B

Please add quantitative and statistical results of immunoblotting from repeated experiments. Please refer to Figures 3, 5 and 6 from the following paper.

Regulation of heat shock protein 27 phosphorylation during microcystin-LR-induced cytoskeletal reorganization in a human liver cell line. Toxicology Letters 207 (2011) 270–277. doi:10.1016/j.toxlet.2011.09.025.

23. Fig. 6BD

Please add quantitative and statistical results of immunoblotting from repeated experiments. Please refer to Figures 3, 5 and 6 from the following paper.

Regulation of heat shock protein 27 phosphorylation during microcystin-LR-induced cytoskeletal reorganization in a human liver cell line. Toxicology Letters 207 (2011) 270–277. doi:10.1016/j.toxlet.2011.09.025.

24. Figure 7

The sizes of words in the figure are too small.

25. Fig. 8A, upper panel, RNA seq data

What do red and green bars mean, respectively?

26. Fig. 8A, lower panel, Fig. 8B

The error bars are not clear, especially for controls and cases that genes were down-regulated. Please re-draw the figures and present data of controls and treatment groups in two parallel bars. Please refer to Fig. 4D.

27. Fig. 8A, lower panel

What is the data type? RNA-Seq or RT-qPCR? What is the relationship between Fig. 8A, lower panel and Fig. 8B?

28. Discussion

The first paragraph

The text is not about results and discussion of this manuscript. Please move this paragraph to the section of “Introduction”.

29. Previous Supplementary Fig. S3

Please add quantitative and statistical results of cell cycle, percentages of cells at different phases of repeated experiments. The data of cell cycle can help understand the status of cells. Please insert the data.

ok

Author Response

Comments and Suggestions for Authors

Journal: Toxins (ISSN 2072-6651)

Manuscript ID: toxins-2372558-peer-review-v2

Type: Article

Title: Cyanotoxins increase cytotoxicity and promote nonalcoholic fatty liver disease progression by enhancing cell steatosis

This manuscript aimed to investigate the effects and impact of low concentrations of MC-LR, MC-RR, NOD, and CYN on hepatotoxicity, hepatic steatosis, and fibrosis in hepatocytes, HepaRG, and HCC cells. The results are interesting. However, the presentation and writing are not good. I have the following comments and suggestions for the authors to improve the quality of manuscript.

  1. Section “Key Contribution”

Lines 23-24

“Exposure to cyanotoxins; microcystin-LR (MC-LR), microcystin-RR (MC-RR), nodularin (NOD), and cylindrospermopsin (CYN) increased liver cell toxicity.”

Please change “;” to “,”.

Response: Changed.

  1. Section “1. Introduction”

Lines 38-39

“People may be exposed to cyanotoxins by drinking or swimming in contaminated water [1].”

People can also be exposed to cyanotoxins by eating contaminated aquatic products. Please cite the following paper. In fact, you have already cited this paper, the present ref. [20]

Challenges of using blooms of Microcystis spp. in animal feeds: A comprehensive review of nutritional, toxicological and microbial health evaluation. Sci Total Environ, 2021. 764: p. 142319.

Response: Done.

  1. Lines 68-70

“Cyanotoxins are divided into cyclic peptides, alkaloids, lipopeptides, non-protein amino acids, and lipoglycans and so far, 279 cyanotoxins have been identified [13].”

Please change to “Cyanotoxins are divided into cyclic peptides, alkaloids, lipopeptides, non-protein amino acids, and lipoglycans. So far, 279 microcystins (MCs) have been identified [13].”

Response: Done.

  1. Lines 80-88

“Although microcystins primarily target the liver, different organs can be impacted such as the gut, kidneys, brain, heart, lungs, and the reproductive system [17-19]. The genus Microcystis contains several species including M. aeruginosa are known to produce several microcystins (MCs) that enhanced toxicity in the kidney, intestine, spleen, liver, and other organs of several fish species [20], and a recent report suggests that cyanotoxins also affect several human organs such as brain, stomach, small and large intestine, lungs, kidneys, skin, and liver, also increased risk of several clinical symptoms [21]. Recent numerous studies indicated that cyanotoxin exposure is associated with hepatic inflammation, gastroenteritis, and liver cancers [18, 19, 22, 23].”

The text is repeated, about muti-organ toxicity of MCs. Please condense the sentences. You can keep the cited references, as they are about different species.

Response: The paragraph was edited in the revised version.

  1. Lines 103-104

“Furthermore, MC-LR-induced nonalcoholic steatohepatitis (NASH) in rats when rats were fed a high-fat or high-cholesterol diet [38].”

Please change “MC-LR-induced” to “MC-LR induced”.

Response: Done.

  1. Section “2. Results”

Lines 128-129

“Although so far about 279 different cyanotoxins have been identified, in the current study we selected four cyanotoxins: MC-LR, MC-RR, NOD, and CYN.”

279 different MCs have been identified. Please delete this sentence.

Response: Done.

  1. Line 156

Please change “nm” to “nM”.

Response: Done.

  1. Lines 164-165

“In SK-Hep1 cells, MC-LR increased cleaved PARP expression (Fig. 1C, middle panel).”

Fig. 1C, the results show no increase of cleaved PARP expression in SK-Hep1 cells. Please check it.

Response:  Text modified. 

  1. Please add statistical results of immunoblotting from repeated experiments. Please refer to Figures 3, 5, and 6 from the following paper.

Regulation of heat shock protein 27 phosphorylation during microcystin-LR-induced cytoskeletal reorganization in a human liver cell line. Toxicology Letters 207 (2011) 270–277. doi:10.1016/j.toxlet.2011.09.025.

Response:  All three time-repeated immunoblots (Fig. 1C, Fig. 2B, Fig. 3B, Fig. 4D&E, Fig. 5B, and  Fig. 6B&D) were quantified by Image J  and statistically presented. The text was modified accordingly in the revised version of the manuscript.

  1. Lines 168-169

“low concentrations of cyanotoxins enhance cytotoxicity”

What do you mean?

Response: The sentence was modified.

  1. Fig. 1B

Please re-analyze the data of HepG2-CYN.

Response: Done

  1. Figures 1-8

Please check and unify capital and small letters of p. For example, p is small in Fig. 1, while is capital in Fig. 5.

Response: All made small “p”.

  1. Line 185

Please delete “and its”.

Response: Done.

  1. Lines 224-226

“As shown in Fig. 3B, MC-LR increased the expression of pERK, and BIP in HepG2 cells and pERK, BIP, peIF2a, and ATF6 in SK-Hep1 cells (Fig. 3B, upper and lower panels).”

Fig. 3B, the results show no increase of pERK expression in HepG2 or SK-Hep1 cells. Please check it.

Response: Text modified. Thanks.

  1. Line 245

Please change “modulates” to “modulate”.

Response: Changed.

  1. Fig. 4A

Please add results of FABP1 in SK-Hep1 cells.

Response: FABP1 expression was not detected in SK-Hep1 cells in our experimental conditions. 

  1. Fig. 4B

Why are there 3 bands for SREBP1?

Response: SREBP shows native (125 kDa) and mature (65 kDa) forms of bands (10.1194/jlr.M200252-JLR200).  A nonspecific band was cropped (Fig. 4D) in the revised manuscript.

  1. Lines 300-302

“Exposure of MC-LR significantly increased mRNA expression of TGFB1, p21 (~ 6 fold), cytokeratin 7 (~4.2 fold), and FGF-23 (~2.2 fold) in HepG2 cells compared with vehicle-treated cells.”

In Fig. 5A, the results show no increase of cytokeratin 7 (~4.2 fold) in HepG2 cells. Please check it.

Response: The text was modified and updated. Thanks.

  1. Figure 7

The sizes of words in the figure are too small.

Response: All panels in Figure 7 were updated and the font size of the axis more than Arial 11 was maintained.

  1. Fig. 8A, upper panel, RNA seq data

What do red and green bars mean, respectively?

Response: Indicated as per suggested and the X axis clearly shows fold change. 

  1. Fig. 8A, lower panel, Fig. 8B

The error bars are not clear, especially for controls and cases that genes were down-regulated. Please re-draw the figures and present data of controls and treatment groups in two parallel bars. Please refer to Fig. 4D.

Response:  Fig. 8A lower panel and Fig. 8 B&C replotted as suggested in the revised manuscript.

  1. Fig. 8A, lower panel

What is the data type? RNA-Seq or RT-qPCR? What is the relationship between Fig. 8A, lower panel and Fig. 8B?

Response:   Fig. 8A upper panel shows RNA-Seq data. Fig 8A lower panels showed validation of RNA-Seq data from hepatocytes after exposure to MC-LR and CYN.  

In Fig. 8 B&C is not RNA seq data. However, we also analyze/validate the expression of these genes (which are differentially expressed in hepatocytes) using HepaRG and HepG2 cell samples after exposure to MC-LR and CYN. The text was modified and mentioned in the revised version of the manuscript. 

  1. Discussion

The first paragraph

The text is not about results and discussion of this manuscript. Please move this paragraph to the section of “Introduction”.

Response: Moved and modified text slightly. 

  1. Previous Supplementary Fig. S3

Please add quantitative and statistical results of cell cycle, percentages of cells at different phases of repeated experiments. The data of cell cycle can help understand the status of cells. Please insert the data.

Response: we already responded that we are going to use this data for future manuscripts.  Thanks.  

Reviewer 2 Report

In general, the authors have taken into account all the comments made and the manuscript has been improved.

This new version of the manuscript can be accepted for publication in the journal TOXINS.

Author Response

Comments and Suggestions for Authors

In general, the authors have taken into account all the comments made and the manuscript has been improved.

This new version of the manuscript can be accepted for publication in the journal TOXINS.

Response: Thanks for accepting our manuscript in Toxins.

Round 3

Reviewer 1 Report

Journal: Toxins (ISSN 2072-6651)

Manuscript ID: toxins-2372558-peer-review-v3

Type: Article

Title: Cyanotoxins increase cytotoxicity and promote nonalcoholic fatty liver disease progression by enhancing cell steatosis

The manuscript "Cyanotoxins increase cytotoxicity and promote nonalcoholic fatty liver disease progression by enhancing cell steatosis " aimed to focusing on effects of low concentrations of MC-LR, MC-RR, NOD, and CYN on hepatotoxicity, hepatic steatosis, and fibrosis in hepatocytes, HepaRG, and HCC cells. The manuscript improved during the revisions. I have read through this manuscript carefully and I list below some points for correction. The most importantly, results of statistical analyses, including one-way ANOVA followed by Tukey HSD post hoc test, for western blot, were not presented. It is meaningless if there are no statistical results presented by * compared with controls. Pleas also insert number of replicates (n) for each experiment.

1. Line 68

There should be full name of DW.

2. Lines 71-75

Please check the layout of the text.

3. Lines 71 & 72

[7]” should not be italic.

4. Line 145

“, nodularin (NOD),” should not be italic.

5. Line 146

liver damage [38]” should not be italic.

6. Line 156

“Oreochromis” should be italic.

7. Line 218

“µM” should be “nM”.

8. Fig. 1CD, bar plots for results of western blot

Please insert results of statistical analyses, including one-way ANOVA followed by Tukey HSD post hoc test. The result should be presented for each protein and each toxin, respectively. Please refer to Fig. 2A or Fig. 4D. It is meaningless if there are no statistical results presented by * compared with controls. Pleas also insert number of replicates (n) for each experiment.

9. Fig. 2B, Fig. 3B, Fig. 4E, Fig. 5B, Fig. 6BD, bar plots

The same comments as Fig. 1CD.

ok

Author Response

Comments and Suggestions for Authors

Journal: Toxins (ISSN 2072-6651)

Manuscript ID: toxins-2372558-peer-review-v3

Type: Article

Title: Cyanotoxins increase cytotoxicity and promote nonalcoholic fatty liver disease progression by enhancing cell steatosis

The manuscript "Cyanotoxins increase cytotoxicity and promote nonalcoholic fatty liver disease progression by enhancing cell steatosis " aimed to focusing on effects of low concentrations of MC-LR, MC-RR, NOD, and CYN on hepatotoxicity, hepatic steatosis, and fibrosis in hepatocytes, HepaRG, and HCC cells. The manuscript improved during the revisions. I have read through this manuscript carefully and I list below some points for correction. The most importantly, results of statistical analyses, including one-way ANOVA followed by Tukey HSD post hoc test, for western blot, were not presented. It is meaningless if there are no statistical results presented by * compared with controls. Pleas also insert number of replicates (n) for each experiment.

 Response: Thanks. Pl see our response below.

  1. Line 68

There should be full name of DW.

Response: Done.

  1. Lines 71-75

Please check the layout of the text.

Response: Adjusted.

  1. Lines 71 & 72

[7]” should not be italic.

Response: Done.

  1. Line 145

“, nodularin (NOD),” should not be italic.

Response: Done.

  1. Line 146

liver damage [38]” should not be italic.

Response: Done.

  1. Line 156

“Oreochromis” should be italic.

Response: Modified.

  1. Line 218

“µM” should be “nM”.

Response: Modified.

  1. Fig. 1CD, bar plots for results of western blot

Please insert results of statistical analyses, including one-way ANOVA followed by Tukey HSD post hoc test. The result should be presented for each protein and each toxin, respectively. Please refer to Fig. 2A or Fig. 4D. It is meaningless if there are no statistical results presented by * compared with controls.

Response: Figures replotted/modified. Differences among multiple means were assessed by one-way ANOVA and Bonferroni post hoc tests with Sigmaplot 14.5 Software (Palo Alto, CA). Since we used low concentrations of cyanotoxins the effect on protein expression is not significant.  We added this information in the revised manuscript and the results were updated. 

  1. Fig. 1A-C, Fig. 2B, Fig. 3B, Fig. 4E, Fig. 5B, Fig. 6BD, bar plots

Pleas also insert number of replicates (n) for each experiment.

Response: Added in figure legends.